# Transductive Conformal Inference for Full Ranking

**Jean-Baptiste Fermanian**
IMAG, IROKO, Univ. Montpellier, Inria, CNRS,
Montpellier, France
`jean-baptiste.fermanian@inria.fr`

**Pierre Humbert**
LPSM, Sorbonne Université,
Paris, France
`humbert@lpsm.paris`

**Gilles Blanchard**
Université Paris Saclay, Institut Mathématique d'Orsay,
Orsay, France
`gilles.blanchard@universite-paris-saclay.fr`

## Abstract

We introduce a method based on Conformal Prediction (CP) to quantify the uncertainty of full ranking algorithms. We focus on a specific scenario where $n + m$ items are to be ranked by some "black box" algorithm. It is assumed that the relative (ground truth) ranking of $n$ of them is known. The objective is then to quantify the error made by the algorithm on the ranks of the $m$ new items among the total $(n + m)$. In such a setting, the true ranks of the $n$ original items in the total $(n + m)$ depend on the (unknown) true ranks of the $m$ new ones. Consequently, we have no direct access to a calibration set to apply a classical CP method. To address this challenge, we propose to construct distribution-free bounds of the unknown conformity scores using recent results on the distribution of conformal p-values. Using these scores upper bounds, we provide valid prediction sets for the rank of any item. We also control the false coverage proportion, a crucial quantity when dealing with multiple prediction sets. Finally, we empirically show on both synthetic and real data the efficiency of our CP method for state-of-the-art ranking algorithms such as RankNet or LambdaMart.

## 1 Introduction

Ranking is a fundamental problem in machine learning where the objective is to sort some items, such as documents or products, by their relevance to a query or user profile. It has a wide range of applications, ranging from document retrieval [Cao et al., 2006], collaborative filtering [Liu and Yang, 2008], sentiment analysis [Liu et al., 2017], and product rating. These applications are grouped under the term *preference learning* [Fürnkranz et al., 2008]. While there exist a lot of ranking algorithms in the literature (see e.g. Liu et al. [2009] for a review), the quantification of the uncertainty in their rank predictions using conformal prediction is relatively new [Angelopoulos et al., 2023a, Xu et al., 2024]. Indeed, so far, the design of ranking algorithms has mainly focused on the training phase and their evaluations regarding the uncertainty of their predicted ranking, is not often studied. With the increasing popularity of ranking methods (especially in the learning-to-rank literature Liu et al., 2009), this evaluation is however a mandatory step to deploy them for real-world applications.

Our main objective is to quantify the uncertainty of ranking algorithms by constructing distribution free prediction set for the ranks of each items. Formally, given $n + m$ items with an underlying unknown total order, we want to construct prediction sets that contain the true rank of each item with high probability. We consider a particular scenario where the first $n$ items have known relative rankings, but their positions within the whole $n + m$ items are unknown. Furthermore, the remaining $m$ items have no known ranking information.

39th Conference on Neural Information Processing Systems (NeurIPS 2025).

This scenario occurs for many problems. In a matchmaking problem for instance [Herbrich et al., 2006, Alman and McKay, 2017, Minka et al., 2018], the $m$ new items are new players entering the game that we want to rank among the previous players to organize matches between players with similar skills. In product rating, the $m$ new items can be new products such as movies that we want to rank among those already known by a particular user for which we have his feedback. In these cases, we can assume that we have access to the exact relative rank of the $n$ first items. We also consider the case where we want to combine an expensive ranking algorithm with a cheaper but less reliable one (based, for example, on a smaller architecture). In this situation, we want to estimate the reliability of the cheaper algorithm compared to the costly one considered as a proxy truth. We can therefore randomly select $n$ items, rank them using the more efficient algorithm, and then use these rankings to quantify the uncertainty of the less efficient algorithm when it ranks the remaining $m$ items.

Formally, for each item $i \in [\![n+m]\!]$, we want to construct a *marginally valid* set which contains, with high probability, its unknown rank $R_i$ inside the $n+m$ items:

$$\mathbb{P}\left( R_i \in \widehat{\mathcal{C}}_i \right) \geqslant 1 - \alpha \,, \tag{1}$$

where $\alpha \in (0,1)$ is a desired miscoverage level. In addition, as we construct prediction sets for multiple points, we also want to control the false coverage proportion, which is the average number of mispredictions. More precisely, we want to construct $m$ prediction sets $(\widehat{\mathcal{C}}_i)_{i=1}^m$ satisfying for some small $\beta > 0$:

$$\mathbb{P}\left( \frac{1}{m} \sum_{i=1}^m \mathbf{1}\{R_i \notin \widehat{\mathcal{C}}_i\} \leqslant \alpha \right) \geqslant 1 - \beta \,. \tag{2}$$

There exist several methods to construct marginally valid sets in regression or classification settings, based on the so-called Conformal Prediction (CP) framework [Papadopoulos et al., 2002, Vovk et al., 2005, Romano et al., 2019], but these do not directly apply in this specific setting of ranking. Indeed, the problem of ranking can be seen as a regression problem where the outcome, the rank, depends on all the points observed, or as a classification problem with a moving number of classes. An important difference with a classical CP framework is that a calibration set is not directly accessible, as knowing the relative rank of the $n$ first items is not sufficient to know their ranks within the whole sample.

**Contributions.** This work constructs marginally valid prediction sets, satisfying Eq. (1), for the ranks of $n+m$ items. We assume already having a ranking algorithm $\mathcal{A}$ (considered as a "black box") and knowing the exact relative ranks of $n$ items. These $n$ items will be used to quantify the uncertainty of the algorithm $\mathcal{A}$. Our main contribution is to propose a general method for this calibration phase. The difficulty is that we cannot directly quantify the algorithm's error on the calibration set, as the true ranks are only known through the relative ranks. We then provide computable bounds of these ranks, supported by theoretical arguments, to construct valid prediction sets for which we also control the false coverage proportion. These findings provide a practical solution to the challenges of constructing valid prediction sets in ranking problems, where traditional CP methods face limitations due to the interdependence of ranks.

## 2 Background

### 2.1 Split conformal prediction

Conformal Prediction (CP) is a framework introduced by Vovk et al. [2005] to construct distribution-free prediction sets quantifying the uncertainty in the predictions of an algorithm. Formally, given a calibration set $\{(X_i, Y_i)\}_{i \in [\![n]\!]}$ of points in $\mathcal{X} \times \mathcal{Y}$ and an algorithm $\mathcal{A}$, a CP method constructs for a new point $X_{n+1} \in \mathcal{X}$ a set $\widehat{\mathcal{C}}$ containing the unobserved outcome $Y_{n+1} \in \mathcal{Y}$ with high probability:

$$\mathbb{P}\left( Y_{n+1} \in \widehat{\mathcal{C}}(X_{n+1}) \right) \geqslant 1 - \alpha \,. \tag{3}$$

In classification, $\mathcal{Y}$ is the set of classes and $\mathcal{A}$ a classifier and in regression, $\mathcal{Y} = \mathbb{R}$ and $\mathcal{A}$ is a regressor. The idea behind CP methods is to first build non-conformity scores $V_i := s(X_i, Y_i)$ on the calibration set, where $s : \mathcal{X} \times \mathcal{Y} \to \mathbb{R}$, is a score function which is intended to quantify the error of the algorithm at a given point. Many score functions have been considered to catch different type of information, see e.g. Angelopoulos et al. [2023b] for a review. In regression, a common choice is to

use the absolute residual $s(x, y) = |y - \mathcal{A}(x)|$ where the algorithm $\mathcal{A}$ can either be trained on an independent sample (split conformal prediction, Papadopoulos et al., 2002) or retrained on a subset of points (full conformal prediction Vovk et al., 2005). In this paper, we will focus on the case where an already trained algorithm is provided, which corresponds to the case of split CP. Then, for this score function and some integer $k$, the prediction set is

$$\widehat{\mathcal{C}}_k(x) := \left\{ y \in \mathcal{Y} \, : \, s(x, y) \leqslant V_{(k)} \right\}, \tag{4}$$

where $V_{(k)}$ is the $k$-th smallest score of $V_1, \ldots, V_n, \infty$. This set is marginally valid under mild assumptions as presented in the following theorem.

**Theorem 2.1** (Vovk et al., 2005, Lei et al., 2018). *If the scores $V_1, \ldots, V_{n+1}$ are exchangeable, then for any $\alpha \in (0, 1)$ and $k = \lceil (1 - \alpha)(n + 1) \rceil$ the set (4) returned by the split CP method satisfies:*

$$\mathbb{P}\left( Y_{n+1} \in \widehat{\mathcal{C}}_k(X_{n+1}) \right) \geqslant 1 - \alpha \, .$$

We refer to Vovk et al. [2005], Angelopoulos et al. [2023b] and Fontana et al. [2023] for in-depth presentations of CP. See also Manokhin [2024] for a curated list of CP papers.

## 2.2 False coverage proportion

Some recent results have considered the transductive setting [Vovk, 2013] where conformal prediction sets are constructed for $m \geqslant 2$ test points $\{(X_{n+i}, Y_{n+i})\}_{i=1}^m$. In this case, while maintaining the coverage guarantee (3), a usual goal is to control the False Coverage Proportion (FCP) defined as:

$$\text{FCP} := \frac{1}{m} \sum_{i=1}^m \mathbf{1}\left\{ Y_{n+i} \notin \widehat{\mathcal{C}}(X_{n+i}) \right\}, \tag{5}$$

where $\widehat{\mathcal{C}}(X_{n+i})$ are some prediction sets. They can be constructed as in Eq. (4) for instance. In words, the FCP is the proportion of observations in the test set that fall outside the constructed prediction sets.

If the prediction sets are of the form (4), under the assumptions of Theorem 2.1 we have $\mathbb{E}[\text{FCP}] \leqslant \alpha$. Furthermore, the exact distribution of the FCP is known [Marques F., 2024, Huang et al., 2024]. Going further, Gazin et al. [2024] studied the full joint distribution of the insertion ranks of test scores into calibration scores, leading under the same assumptions to a uniform control of the FCP. The asymptotic behaviour of the FCP in the form of a functional CLT has also been studied recently by Gazin [2024]. With all these theoretical results, it is possible to control in probability the FCP by adjusting the coverage of the prediction sets.

## 2.3 Related work in ranking

Given a set of items the goal of ranking is to infer a particular ordering over these items. There exist a multitude of settings and approaches to solve the ranking problem.

In the learning-to-rank literature, and more specifically, in the pairwise approach, the ranking task is formalized as a classification of pairs of items into two classes: a tuple is correctly ranked or incorrectly ranked. Herbrich et al. [1999] proposed an algorithm based on Support Vector Machine (SVM) called RankSVM to solve this classification problem. Alternative techniques can be used such as boosting with RankBoost [Freund et al., 2003], Neural Network with RankNet [Burges et al., 2005], random forest with LambdaRank [Burges et al., 2006] and LambdaMart [Wu et al., 2010] (see Burges, 2010, Qin et al., 2021 for an overview). Other methods based on the pointwise or listwise approaches are also widely study [Cao et al., 2007, Liu et al., 2009, Li, 2011].

Another line of research solve the ranking problem through binary comparisons of items. The different settings considered allow these binary comparisons to be measured either completely at random [Wauthier et al., 2013], actively [Ailon, 2012, Heckel et al., 2019], or several times [Negahban et al., 2012, Feige et al., 1994]. They can also assume that all the possible comparisons between the items are known but up to some noise [Braverman and Mossel, 2007, 2009]. A significant body of literature also focuses on the problem of uncertainty quantification in ranking problems under the Bradley-Terry-Luce model [Bradley and Terry, 1952, Luce et al., 1959]. For instance, Liu et al.

[2023] consider the BTL model and infer its general ranking properties. Gao et al. [2023] and Fan et al. [2025] study the maximum likelihood estimator and the spectral estimator to estimate the parameters of the BTL model, enabling them to construct confidence intervals for individual ranks.

However, the quantification of uncertainty of these ranking algorithms has been little studied in a CP framework. Recently, Angelopoulos et al. [2023a] have developed a CP method for quantifying the uncertainty of learning-to-rank algorithms specially tailored for recommendation systems. Xu et al. [2024] also proposed to use CP, but for the ranked retrieval task. While related to the present work, they do not consider the same setting.

## 3    Conformal prediction for ranking algorithms

We now describe our setting, as well as our methodology to provide valid prediction sets.

**Notation:** For a point $y$ in $\mathbb{R}$ and a finite subset $\mathcal{D} \subset \mathbb{R}$, the rank of $y$ in $\mathcal{D}$ is $\mathtt{R}(y, \mathcal{D}) := \sum_{z \in \mathcal{D}} \mathbf{1}\{y \geqslant z\}$. For an index $1 \leqslant i \leqslant |\mathcal{D}|$, if $\mathcal{D}$ has no ties, we denote the point of $\mathcal{D}$ of rank $i$ by $\mathtt{R}^{-1}(i, \mathcal{D}) := \sum_{z \in \mathcal{D}} z \mathbf{1}\{\mathtt{R}(z, \mathcal{D}) = i\}$. A multiset (or bag) will be denoted $\wr \cdot \wr$ and is an unordered set with potentially repeated elements.

### 3.1    Setting

Let us consider $n + m$ items, each associated with a pair $(X_i, Y_i) \in \mathcal{X} \times \mathbb{R}$. We suppose that these items are divided into two sets: a calibration set of size $n$, $\{(X_i, Y_i)\}_{i \in [\![n]\!]}$ and a test set of size $m$, $\{(X_{n+i}, Y_{n+i})\}_{i \in [\![m]\!]}$. For any item $i$, $X_i$ is, for instance, a vector of features, while $Y_i$ allows us to properly define its true rank within each subset or the entire set through the following variables:

$$R_i^c = \mathtt{R}\big(Y_i, \wr Y_j \wr_{j \in [\![n]\!]}\big), \ \ R_i^t = \mathtt{R}\big(Y_i, \wr Y_{n+j} \wr_{j \in [\![m]\!]}\big), \ \ R_i^{c+t} = \mathtt{R}\big(Y_i, \wr Y_j \wr_{j \in [\![n+m]\!]}\big) = R_i^c + R_i^t \ .$$

We assume throughout the paper that:

**Assumption 3.1.** The vector $(Y_i)_{i \in [\![n+m]\!]}$ is *exchangeable* and has *no ties*. In particular, this implies that there is a *total order* between the $n + m$ items.

The values $(Y_i)_i$ can be interpreted as an *unobservable underlying truth*. In our setting, we insist that only the features $(X_i)_{i \in [\![n+m]\!]}$ and the true ranks of the calibration points $(R_i^c)_{i \in [\![n]\!]}$ are observed. For example, they can model the intrinsic skill of some players only observable by comparing them with each other. Assumption 1 is needed to model correctly our problem of full ranking. The following assumption is essential to be able to construct conformal prediction sets for the ranks.

**Assumption 3.2.** The vector $(X_i, R_i^{c+t})_{i \in [\![n+m]\!]}$ is exchangeable.

To estimate the true ranks $R_1^{c+t}, \ldots, R_{n+m}^{c+t}$, we assume that we have access to an algorithm $\mathcal{A} : \mathcal{X} \times \mathcal{X}^{n+m} \to \mathbb{K}$ which is intended to predict the rank of a point inside a set of points. Its inputs are the target item and the multiset of items among which it seeks to sort it. For clarity, when there is no confusion, we will sometimes omit the dependence in the multiset of items: $\mathcal{A}(x) := \mathcal{A}(x, \wr x_\ell \wr_\ell)$. This algorithm can predict for each point either a *rank* or a *value* from which a rank can be deduced. For these two situations, we propose two conformity score functions which quantify the error made by $\mathcal{A}$ in its ranking:

**(RA) setting:** $\mathbb{K} = \mathbb{N}$, the rank is directly predicted by the algorithm $\mathcal{A}$: $\widehat{R}_i^{c+t} := \mathcal{A}\big(X_i, \wr X_j \wr_{j \in [\![n+m]\!]}\big)$. In this situation, we can consider the classical residual scores, i.e., for $x, (x_j)_j$ in $\mathcal{X}$ and $r \in \mathbb{N}$:

$$s^{\mathbf{RA}}(x, r, \wr x_j \wr_j) := |r - \mathcal{A}(x, \wr x_j \wr_j)| \ .$$

This is simply the absolute difference between $r$ and the predicted rank of $x$ inside $(x_j)_j$.

**(VA) setting:** $\mathbb{K} = \mathbb{R}$, the predicted rank is deduced from values constructed by the algorithm $\mathcal{A}$:

$$\widehat{R}_i^{c+t} := \mathtt{R}\Big(\mathcal{A}\big(X_i, \wr X_\ell \wr_\ell\big), \Big\{\mathcal{A}\big(X_j, \wr X_\ell \wr_\ell\big)\Big\}_{j \in [\![n+m]\!]}\Big).$$

Here, because $\mathcal{A}(x)$ is a real value, we consider a more refined score function than $s^{\mathbf{RA}}$, for $x$, $(x_j)_j$ in $\mathcal{X}$ and $r \in \mathbb{N}^*$:

$$s^{\mathbf{VA}}(x, r, \langle x_j \rangle_j) := \left| \mathtt{R}^{-1}\left(r, \langle \mathcal{A}(x_j) \rangle_j\right) - \mathcal{A}(x) \right|.$$

This score quantifies the distance between the value attributed by the algorithm to the point $x$ and the value it should have had to be at rank $r$. We restrict ourselves to these two scores for our experiments, but our methodology and theoretical results are more generally valid for any score function $s(x, r, \langle x_i \rangle_i)$.

*Remark* 3.3. The fact that the algorithm depends on the multiset rather than an $(n+m)$-tuple is important, as it implies that the algorithm treats the points exchangeably. With Assumption 3.2, the vector $(\mathcal{A}(X_i))_{i \in [\![n+m]\!]}$ is thus exchangeable.

*Remark* 3.4. By considering the predicted ranks as scores, we see that (**RA**) is a sub-case of (**VA**). Moreover, as in (**RA**), $\mathtt{R}^{-1}(r, \langle \mathcal{A}(x_i) \rangle_i)_i = r$, $s^{\mathbf{VA}}$ reduces to the residual function $s^{\mathbf{RA}}$.

### 3.2 Methodology and main results

We now introduce our main theoretical results to construct prediction sets for the ranks $R_1^{c+t}, \ldots, R_{n+m}^{c+t}$. Recall that our objective is to build for $j \in [\![n+m]\!]$ a marginally valid set $\widehat{\mathcal{C}}_j$ for $R_j^{c+t}$, i.e. satisfying Eq. (1) for a given $\alpha \in (0, 1)$. As we construct multiple prediction sets, we also provide a control of the FCP (Eq.(5)).

To construct these sets, a natural strategy is to calculate the quantile of order $1 - \alpha$ of the scores $s(X_i, R_i^{c+t}, \langle \mathcal{A}(X_j) \rangle_{j \in [\![n+m]\!]})$ computed on the calibration set as this is done in the split CP method (see Section 2.1). However, these scores depend on the quantities $\left\{ R_i^{c+t} \right\}_{i \in [\![n]\!]}$ which are unknown. Consequently, they are not computable and we must find another strategy. The following result shows that, if we can bound the ranks of the calibration $\left( R_i^{c+t} \right)_{i \in [\![n]\!]}$ with high probability, then it is possible to construct marginally valid sets for the test ranks $\left\{ R_{n+i}^{c+t} \right\}_{i \in [\![m]\!]}$.

**Theorem 3.5.** *Let* $\alpha \in (0, 1)$, $\delta < \alpha$ *and* $R_i^-, R_i^+ \in \mathbb{R}$ ($i \in [\![n]\!]$) *be random variables depending on* $(X_i, R_i^c)_{i \in [\![n]\!]}$ *and satisfying*

$$\mathbb{P}\left(\forall i \in [\![n]\!] : R_i^{c+t} \in \left[ R_i^-, R_i^+ \right]\right) \geqslant 1 - \delta. \tag{6}$$

*Let us define the proxy scores for* $i \in [\![n]\!]$:

$$S_i := \max_{r \in \left[R_i^-, R_i^+\right]} s\left(X_i, r, \langle X_j \rangle_{j \in [\![n+m]\!]}\right) \tag{7}$$

*and the associated conformal prediction set:*

$$\widehat{\mathcal{C}}_k(x) = \left\{ r : s\left(x, r, \langle X_i \rangle_{i \in [\![n+m]\!]}\right) \leqslant S_{(k)} \right\}, \tag{8}$$

$S_{(k)}$ *being the $k$-th smallest proxy score in* $S_1, \ldots, S_n$.

*If Assumption 3.1 is satisfied, then for* $k = \lceil (1 - \alpha + \delta)(n+1) \rceil$ *and any* $i \in [\![m]\!]$:

$$\mathbb{P}\left(R_{n+i}^{c+t} \in \widehat{\mathcal{C}}_k(X_{n+i})\right) \geqslant 1 - \alpha.$$

For the score functions $s^{\mathbf{RA}}$ and $s^{\mathbf{VA}}$, the proxy scores of Eq. (7) and the associated conformal sets have explicit expressions which make them easy to compute.

*Example* 3.6 ($s = s^{\mathbf{RA}}$). The maximum (7) is necessarily attained at the edges of the interval $[R^-, R^+]$:

$$\max_{r \in [R^-, R^+]} s^{\mathbf{RA}}(x, r, \langle x_i \rangle_i) = \max\left( \left| R^- - \mathcal{A}(x, \langle x_i \rangle_i) \right|, \left| R^+ - \mathcal{A}(x, \langle x_i \rangle_i) \right| \right).$$

The associated conformal set is $\widehat{\mathcal{C}}^{\mathbf{RA}}(x) = \left[ \mathcal{A}(x, \langle X_i \rangle_i) \pm S_{(k)} \right]$.

*Example* 3.7 ($s = s^{\mathbf{VA}}$). Similarly as the previous example, the maximum is attained at the edges of $[R^-, R^+]$:

$$\max_{r \in [R^-, R^+]} s^{\mathbf{VA}}(x, r, \langle x_i \rangle_i) = \max_{r \in \{R^-, R^+\}} \left( \left| \mathcal{A}(x) - \mathtt{R}^{-1}(r, \langle \mathcal{A}(x_i) \rangle_i) \right| \right).$$

The corresponding conformal set is then the interval

$$\widehat{\mathcal{C}}^{\mathbf{VA}}(x) = \left[ \mathtt{R}\Big(\mathcal{A}(x) - S_{(k)}, \wr\mathcal{A}(x_i)\wr_i\Big), \mathtt{R}\Big(\mathcal{A}(x) + S_{(k)}, \wr\mathcal{A}(x_i)\wr_i\Big) \right].$$

We see that contrary to the conformal intervals $\widehat{\mathcal{C}}^{\mathbf{RA}}$ which have the same length for any point $x$, the length of the conformal interval $\widehat{\mathcal{C}}^{\mathbf{VA}}$ can vary depending on the amount of points around $\mathcal{A}(x)$.

The following result gives a control in high probability of the false coverage rate on the test sample.

**Proposition 3.8.** *Let $\alpha \in (0,1)$ and consider the prediction sets $\widehat{\mathcal{C}}_k$ defined in Eq. (8). Assume Assumption 3.2 holds, let $k = \lceil (1-\alpha)(n+1) \rceil$, then with probability at least $1 - \beta - \delta$:*

$$\frac{1}{m} \sum_{i=1}^{m} \mathbf{1}\Big\{ R_{n+i}^{c+t} \notin \widehat{\mathcal{C}}_k(X_{n+i}) \Big\} \leqslant \alpha + \lambda_{n,m} , \tag{9}$$

*where $\lambda_{n,m} = \sqrt{\dfrac{\log(C\sqrt{\tau_{n,m}}/\beta)}{\tau_{n,m}}}$, $\tau_{n,m} = \dfrac{nm}{n+m}$ and $C = 4\sqrt{2\pi}$ works.*

When the sample sizes increase, then $\lambda_{n,m} \to 0$ and the average error on the test sample is at most $\alpha$. In practice, using $\alpha' = \alpha - \lambda_{n,m}$ to control the FCP at level $\alpha$ is too conservative. We instead use the numerical procedure explained by Gazin et al. [2024, Remark 2.6] to find a sharper but implicit value for $\lambda_{n,m}$. They use the exact distribution of the FCP to adjust the level of each set accordingly, in order to control it (see Appendix C for details).

## 3.3 Alternative targets

In the previous section, we focused on the construction of prediction sets for the ranks of test points within the entire data set. In this section, we discuss how to derive from this initial construction, predictions sets for alternative targets. More details are provided in Appendix B.

**Calibration points.** To obtain predictions sets for the calibration points, we can directly use the high probability bounds (6) on which our method is built. Indeed, by construction the sets $[R_i^-, R_i^+]$ contain $R_i^{c+t}$ and have a null FCP with probability at least $1 - \delta$ for $i \in [\![n]\!]$.

**Rank among the test points.** A user might also be interested in the ranking of the test points among themselves, rather than within the entire data set. Such sets can be constructed directly from the previous sets (8), by substracting the number of calibration points of smaller rank to each interval boundary (see Corollary B.1).

**Top-$\mathtt{K}$ items.** It is also possible to target the top-$\mathtt{K}$ items by selecting all items whose associated prediction set includes at least one rank smaller than or equal to $\mathtt{K}$. This strategy can be sufficient if we target a "top" proportion of the items (see Corollary B.2).

## 4 Rank of the calibration points

To make our method complete, it remains to find $R_i^-$ and $R_i^+$ satisfying Eq. (6). Let us recall that $R_i^{c+t} = R_i^c + R_i^t$. Hence, we know that $R_i^{c+t} \in [R_i^c, R_i^c + m]$ and we can use Theorem 3.5 with $R_i^- = R_i^c$ and $R_i^+ = R_i^c + m$. However, using these bounds for instance with the residual conformity function $s^{\mathbf{RA}}$, will increase the length of the conformal set $\widehat{\mathcal{C}}^{\mathbf{RA}}(x)$ by $m$. In this section, we show that it is possible to find theoretical bounds and practical bounds satisfying Eq. (6) that do not excessively increase the length of the final conformal sets.

### 4.1 Theoretical bound

As the calibration and test values $(Y_i)_{i \in [\![n+m]\!]}$ are exchangeable (Assumption 3.1), we can expect that a calibration point with a low rank in the calibration set will have a low rank in the test set. The following proposition justifies this intuition.

**Proposition 4.1.** *Under Assumption 1, for $\delta \in (0,1)$:*

$$\mathbb{P}\Big( R_i^{c+t} \in \big[ R_i^-, R_i^+ \big] \cap [\![n+m]\!], i \in [\![n]\!] \Big) \geqslant 1 - \delta ,$$

*where* $R_i^\pm = R_i^c + (m+1)\left(\dfrac{R_i^c}{n} \pm \sqrt{\dfrac{\log(C\sqrt{\tau_{n,m}}/\delta)}{\tau_{n,m}}}\right)$, $\tau_{n,m} = \dfrac{nm}{n+m}$, *and* $C = 4\sqrt{2\pi}$ *works.*

The proof can be found in the supplementary and relies on Gazin et al. [2024, Theorem 2.4]. While the naive method provides an interval of length $m$ for the rank $R^{c+t}$, our approach achieves an interval of length $O(m/\sqrt{\tau_{n,m}})$. The length has been thus reduced by a factor $\sqrt{\tau_{n,m}} \simeq \min(\sqrt{n}, \sqrt{m})$.

This explicit theoretical bound provides qualitative insight into the behavior and evolution of the ranks with $(m, n)$. However, the constant $C$ is not tight (see Figure 1). To obtain finer envelopes, we propose in the following a numerical method to calibrate them, with theoretical guarantee (Proposition 4.2).

## 4.2 Numerical bounds

The explicit bounds given by Proposition 4.1 are often too conservative in practice. In this section, we construct sharp numerical bounds $R_i^-, R_i^+$ for the ranks $R_i^{c+t}$ such that the event

$$\left\{\forall\, 1 \leqslant j \leqslant n,\ R_j^{c+t} \text{ is between } R_i^- \text{ and } R_i^+\right\}$$

holds with high probability. Let us denote for $r \in [\![n]\!]$, $R_{(r)}^{c+t}$ the rank in the whole sample of the item of rank $r$ in the calibration. In particular, $R_{(R_i^c)}^{c+t} = R_i^{c+t}$. Our idea is to use that the random vector $R_{srt}^{c+t} = (R_{(1)}^{c+t}, \ldots, R_{(n)}^{c+t})$ follows a universal distribution independent of the distribution of the data to simulate them and to estimate an envelope.

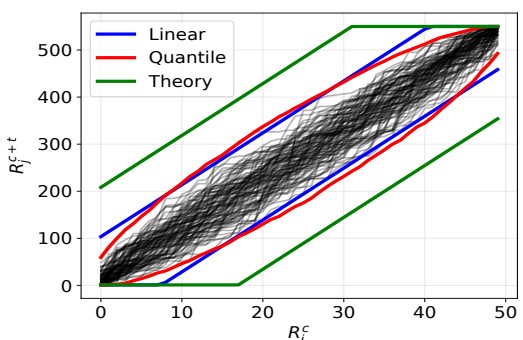

Figure 1: The two envelopes for $n = 50$, $m = 500$, $\delta = 0.1$ and $K = 10^5$. The blue and red lines are respectively the linear and quantile envelopes, and each black curve is an ordered realization of $R^{c+t}$. The green line is the envelope from Prop. 4.1.

This distribution can be drawn using Algorithm 1 thanks to the exchangeability of the data (Assumption 3.1; see Gazin et al., 2024 for more details). An envelope can then be estimated by a Monte-Carlo approach using the empirical cumulative distribution function. We consider two forms of envelope in this work, the linear and the quantile detailed below, which are parameterized by real factors $c$ and $\gamma$. To estimate these

---

**Algorithm 1** Simulation of $R_{srt}^{c+t}$

---

1: Draw $n + m$ uniform random variable $U_i$ on $[0, 1]$.
2: **for** $i = 1, \ldots, n$ **do**
3: $\quad \tilde{R}_i^{c+t} \leftarrow \text{R}(U_i, \{U_j\}_{j \in [\![n+m]\!]})$
4: **end for**
5: **Output:** Sort $\tilde{R}^{c+t}$

---

envelopes, we draw $K$ realizations of vectors following the same distribution as $\left(R_{srt}^{c+t}\right)$ by repeating $K$ times Algorithm 1 and estimate the (multidimensional) cumulative function of this sample. Then, we optimize the parameter $c$ or $\gamma$ for minimizing the width of the envelope (which depends directly on this parameter) while containing a proportion $1 - \delta$ of the $K$ trajectories.

**Linear envelope:** We consider an envelope of the form of Proposition 4.1:

$$\widehat{R}_i^\pm = R_i^c + (m+1)\left(\frac{R_i^c}{n} \pm \widehat{c}\right),$$

where $\widehat{c}$ is minimized under the condition that a proportion $(1 - \delta)$ of the generated vectors are in the envelope. We then take the minimum with $n + m$ and the maximum with $1$ to keep the bounds in $[\![n+m]\!]$.

**Quantile envelope:** For an item of rank $r = R_i^c$, we propose to choose the bound $\widehat{R}_i^+$ (respectively $\widehat{R}_i^-$) as the empirical quantile of order $1 - \widehat{\gamma}$ (resp. $\widehat{\gamma}$) of the simulated ranks $R_{(r)}^{c+t}$, i.e. the ranks in the whole sample of the points of rank $r$ in the calibration. As we want uniform control of these bounds, we need to adjust $\gamma$. Hence, the parameter $\gamma$ is maximized under the condition that a proportion $1 - \delta$ of the generated vectors are in the envelope. Algorithm 2 in Appendix C gives the full process to construct this envelope.

As it can be observed in Figure 1, the quantile envelope is smaller for the low and high rank points while the linear envelope is smaller for the points of medium rank. Furthermore, they both outperform the theoretical envelope.

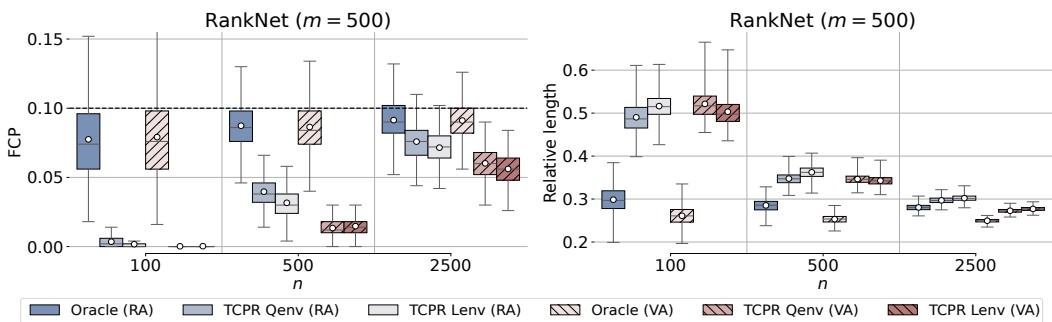

Figure 2: Synthetic data: FCP and relative lengths obtained for RankNet with the (**RA**) and (**VA**) score, for the quantile (*Qenv*) and linear (*Lenv*) envelopes when $m = 500$ and $n \in \{100, 500, 2500\}$. White circles represent the means.

**Monte-Carlo guarantees:** The estimation of the envelopes brings another source of error in our procedure. In the following proposition, we show that this error can be easily controlled and the parameter $K$ can be chosen large enough to make it negligible.

**Proposition 4.2.** *Let $K, n, m \in \mathbb{N}^*$ and $(R^{(k)})_{k \in [\![K]\!]}$ a $K$-sample of ordered ranks drawn from Algorithm 1 with parameters $n$ and $m$. If $(\widehat{R}_i^{\pm})_{i \in [\![n]\!]}$ is an estimated envelope, possibly depending on $(R^{(k)})_k$ such that, almost surely $\frac{1}{K} \sum_{k=1}^{K} \mathbf{1}\left\{\exists i : R_i^{(k)} \notin \left[\widehat{R}_i^-, \widehat{R}_i^+\right]\right\} \leqslant \delta$, then:*

$$\mathbb{P}\left[\forall i : R_i^{c+t} \in \left[\widehat{R}_{R_i^c}^-, \widehat{R}_{R_i^c}^+\right]\right] \geqslant 1 - \delta - 4\sqrt{\frac{\log nK}{K}} \, . \tag{10}$$

The proof is given in Appendix A.5. Notice that this result is not specific to the linear or the quantile envelope we consider but rather to the Monte-Carlo procedure itself. It is indeed based on a control of the multi-dimensional empirical cumulative distribution function from Naaman [2021].

## 5 Experiments

In this section, we empirically evaluate the performance of our CP method referred to as Transductive CP for Ranking (TCPR) on different datasets and algorithms. We construct prediction sets using the score function $s^{\mathbf{RA}}$ or $s^{\mathbf{VA}}$ and with the linear or the quantile envelopes described in Section 4.2. We compare ourselves to the *Oracle* method, which is the conformal method using non-conformity scores calculated with the unobserved true ranks $\left\{R_i^{c+t}\right\}_{i \in [\![n]\!]}$. This comparison allows for an evaluation of the effect of the envelope.

The code of our method is available at `https://github.com/pierreHmbt/transductive-conformal-inference-for-ranking`.

**Ranking algorithms:** We use in the experiments the following algorithms: RankNet [Burges et al., 2005], LambdaMART [Wu et al., 2010] and RankSVM [Herbrich et al., 1999]. These methods learn a score function of the features from pairwise comparisons. In the Appendix, we also use the Balanced Rank Estimation (BRE) [Wauthier et al., 2013] which ranks items using a limited number of comparisons.

**Parameters:** The parameters $\alpha$, $1 - \beta$ and $\delta$, equal to respectively, the probability of miscoverage, the probability to control the FCP at level $\alpha$, and the probability of the quantile envelope, are set to, respectively, $0.1$, $0.75$ and $0.02$, in all our experiments.

**Metrics:** We use the following metrics to evaluate our method. **1)** The FCP of the new ranks (Eq (5)). **2)** The *relative length* of the prediction sets defined as the average size of the sets divided by $n + m$. **3)** The *oracle ratio* defined as the ratio between the size of the set from TCPR and from the oracle.

Some metrics are not presented in this section and can be found in Appendix D with additional information on the data sets and the algorithms, such as their hyperparameters and architecture.

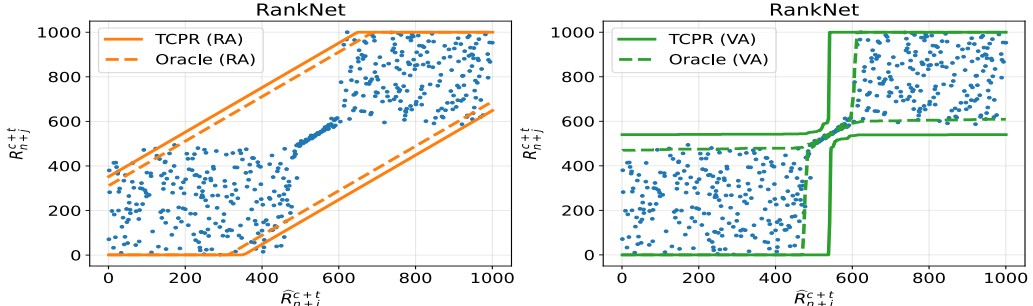

Figure 3: Data from Section 5.1: True ranks $R_{n+j}^{c+t}$ in function of their predicted rank $\widehat{R}_{n+j}^{c+t}$ by RankNet and their prediction sets with scores $s^{\mathbf{RA}}$ and $s^{\mathbf{VA}}$ for $n = m = 500$.

## 5.1 Synthetic data

**Data generation:** We use the same generation process as Pedregosa et al. [2012] with slight modifications to match our setting. In detail, we draw $n + m$ pair of points $(X_i, Y_i)$ where $X_i$ is a Gaussian vector $\mathcal{N}(0, I_d)$ of dimension $d = 5$, $Y_i = \left(1 + \exp(-w^T X_i)\right)^{-1} + \varepsilon_i$ with $w \sim \mathcal{N}(0, I_d)$ and $\varepsilon_i \sim \mathcal{N}(0, 0.07)$. The rank $R_i$ of an observation is defined as the rank of the corresponding $Y_i$. The sizes of the calibration and test sets are chosen among $n, m \in \{100, 500, 2500\}$ and for each pair $(n, m)$, we compute the FCP and the relative length on $B = 1000$ generated data sets.

**Results:** Figure 2 displays the FCP and the relative length for RankNet with the score function $s^{\mathbf{RA}}$ or $s^{\mathbf{VA}}$ and the two proposed envelopes when $m = 500$. Results for other values of $n$ and $m$ and for LambdaMART, RankSVM and BRE are in Appendix D.

We first remark that the FCPs are correctly controlled by our method as they remain below $\alpha = 0.1$ at least $100 \cdot (1 - \beta)\% = 75\%$ of the time, as indicated by the upper boundary of the boxes. It is important to note, however, that due to the necessity of using proxy scores, our method tends to be conservative for small values of $n$. For example, when $n = 100$, we see that the FCPs obtained with TCPR are close to $0$ whereas those obtained with the oracle method are in average near $0.07$. Nevertheless, this conservatism diminishes as the calibration size $n$ increases with FCPs close to $0.1$ when $n = 2500$. In terms of relative length, the prediction sets constructed using the quantile envelope are, on average, consistently smaller than those obtained with the linear envelope for FCPs closer to $0.1$. The quantile envelope is therefore preferable. Finally, notice that, at least for RankNet, using $s^{\mathbf{RA}}$ or $s^{\mathbf{VA}}$ gives sets with similar sizes.

**Adaptivity of score $s^{\mathbf{VA}}$.** To further highlight the adaptability of score $s^{\mathbf{VA}}$ relatively to $s^{\mathbf{RA}}$, we consider another data set of size $n = m = 500$. Each pair $(X_i, Y_i)$ is defined by $Y_i = X_i + \varepsilon_i$ where $X_i \sim \text{Beta}(.04, .04)$ and $\varepsilon_i \sim \mathcal{N}(0, .07)$. Here, because the random variables $(X_i)_i$ follow a Beta distribution with parameters lower than $1$, the probability of observing a value near $0$ or $1$ is much larger than the probability of observing a value near $0.5$. As there is more observations close to these values, RankNet ranks with more difficulty the observations near $0$ or $1$ than the others. However, as the prediction sets constructed with $s^{\mathbf{RA}}$ have all the same size, they do not reflect this difficulty (see an example in Figure 3 left panel). On the contrary, the sets returned with $s^{\mathbf{VA}}$ are narrower when the ranking is easier (close to $500 = 0.5 \cdot (n + m)$) and wider for small and large ranks (Figure 3 right panel).

## 5.2 Real data: Yummly-10k

**Dataset:** We evaluate our approach on the Yummly Food-10k data set which consists in $12624$ images of dishes embedded in $\mathbb{R}^{101}$. These embeddings have been constructed to reflect similarities in taste among the dishes (see Wilber et al., 2015[1] for a complete description).

**Ranking task:** As in Canal et al. [2019], to define a full order of these items, we select an (unknown) preference point $x^*$ and define this ordering using the distance to this point. The $(Y_i)_i$ are thus these distances. We use RankNet, LambdaMART, and RankSVM to rank the dishes and our method

---
[1]Companion website: http://vision.cornell.edu/se3/projects/concept-embeddings

| | Rank error | FCP (**RA**) | RL (**RA**) | OR (**RA**) | FCP (**VA**) | RL (**VA**) | OR (**VA**) |
|---|---|---|---|---|---|---|---|
| **RankNet** | 675.9 | $0.069 \pm 0.005$ | $0.306 \pm 0.003$ | $1.095 \pm 0.003$ | $0.069 \pm 0.005$ | $0.312 \pm 0.003$ | $1.104 \pm 0.005$ |
| **LambdaMART** | 714.8 | $0.072 \pm 0.004$ | $0.324 \pm 0.003$ | $1.087 \pm 0.005$ | $0.071 \pm 0.005$ | $0.323 \pm 0.003$ | $1.089 \pm 0.004$ |
| **RankSVM** | 751.3 | $0.076 \pm 0.005$ | $0.400 \pm 0.004$ | $1.064 \pm 0.003$ | $0.071 \pm 0.004$ | $0.408 \pm 0.004$ | $1.079 \pm 0.005$ |

Table 1: Results on the Yummly-10k data set. RL is for Relative Length and OR for Oracle Ratio.

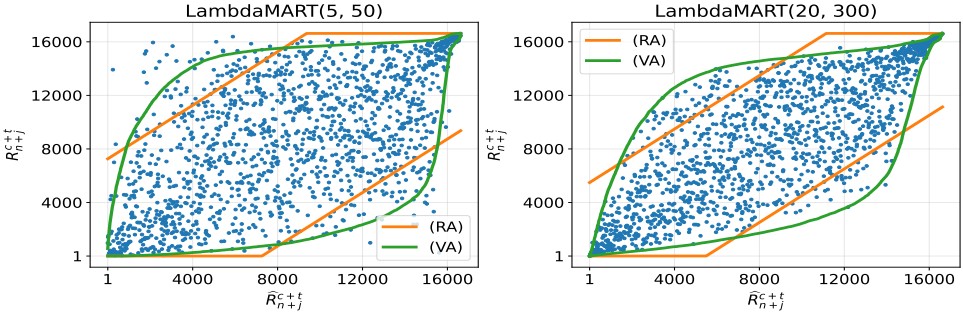

Figure 4: Ranks predicted by LambdaMART(leaves, trees) in function of the ranks predicted by a larger model of $400$ trees of $20$ leaves.

TCPR with the quantile envelope to construct the prediction sets. We divide the data into a training, calibration and test sets of respective size $n_{tr} = 2624$, $n = 2000$ and $m = 8000$. We repeat this splitting $10$ times to compute the metrics.

**Results:** Table 1 reports the values of the different metrics. Firstly, all methods are well-calibrated with FCP consistently remaining below the chosen threshold $0.1$. Secondly, the size of the intervals decreases as performance improves, as reflected by the rank error and the relative length metrics. The impact of the envelope is reflected in the ratio with the oracle. This impact is negligible as the ratio remains close to $1$. We can however notice that it increases as model performance improves. Indeed, the more effective the algorithm, the more the error introduced by the envelope becomes noticeable and impacts the size of the prediction set. Additional results are given in Appendix E.1.

### 5.3 Real data: Anime recommendation LTR data set

**Dataset and ranking task:** This dataset is composed of features of $16681$ movies, characteristics of $15163$ users and $10^6$ ratings associated with a tuple (user, movie) with values ranging from $0$ to $10$. Given the characteristics of a new user, the objective is to produce an ordered list of all the movies ranked by the user's level of interest. We aim to quantify the uncertainty of a smaller model relative to the performance of a larger one, which serves as a reference and defines the ground-truth full ordering of the items, i.e., $R_i^{c+t}$. More details are provided in Appendix E.2.

**Results:** As shown in Figure 4, the size of the prediction sets for both (**RA**) and (**VA**) decreases as the model quality improves. In particular, LambdaMART (leaves=20, trees=300) yields results that are closer to those of the larger model (LambdaMART with 20 leaves and 400 trees) than the smaller ranker (LambdaMART with 5 leaves and 50 trees). This approach has the advantage of allowing for an easy comparison between the two methods without requiring the large model to be run on all movies which can be expensive. We can also point out again the better adaptivity of the score $s^{\textbf{VA}}$ The metrics and comparisons with other architectural configurations are presented in Appendix E.2.

## 6 Conclusion

We have developed a conformal prediction method that quantifies the error of an algorithm ranking $m$ new items among $n$ previously ranked items. Our approach is based on constructing an envelope around these $n$ ranks to quantify the algorithm's error. Both theoretical and empirical evidence demonstrate that this envelope does not significantly impact the size of the intervals, especially when $n$ is large relative to $m$. One limitation of our work is that we only focus on full ranking. It would therefore be interesting to extend our method to a partial ranking framework. Another important line of research would be to cover top-$k$ algorithms. Finally, the dependence of our method on the choice of the envelope we select should be investigated. We expect that a choice more adapted to the problem considered could improve the size of the prediction sets.

## Acknowledgements

The authors acknowledge Ulysse Gazin and Etienne Roquain for helpful comments. G.B. gratefully acknowledges funding from the grants ANR-21-CE23-0035 (ASCAI), ANR-19-CHIA-0021-01 (BISCOTTE) and ANR-23-CE40-0018-01 (BACKUP) of the French National Research Agency ANR. P.H. gratefully acknowledges the Emergence project MARS of Sorbonne Université.

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

# Appendix

## A  Proofs

### A.1  Proof of Theorem 3.5

Let us denote $E_\delta = \left\{ \forall i \in [\![n]\!] : R_i^{c+t} \in \left[ R_i^-, R_i^+ \right] \right\}$. On the event $E_\delta$, for all $i \in [\![n]\!]$, we have:

$$S_i^{\texttt{true}} := s\big(X_i, R_i^{c+t}, (X_k)_{k \in [\![n+m]\!]}\big) \leqslant \max_{r \in [R_i^-, R_i^+]} s\big(X_i, r, (X_k)_{k \in [\![n+m]\!]}\big) := S_i \,.$$

Hence, the quantiles of the "true" scores $S_i^{\texttt{true}}$ are upper bounded by the quantiles of the proxy scores $\{S_i\}_{i \in [\![n]\!]}$, i.e., for all $k \in [\![n]\!]$:

$$S_{(k)}^{\texttt{true}} \leqslant S_{(k)} \,. \tag{11}$$

Then, for $i \in [\![m]\!]$:

$$
\begin{aligned}
\mathbb{P}\Big( R_{n+i}^{c+t} \notin \widehat{\mathcal{C}}(X_{n+i}) \Big) &\leqslant \mathbb{P}\Big( R_{n+i}^{c+t} \notin \widehat{\mathcal{C}}(X_{n+i}) \cap E_\delta \Big) + \mathbb{P}(E_\delta^c) \\
&\leqslant \mathbb{P}\big( s\big(X_{n+i}, R_{n+i}^{c+t}, (X_j)_{j \in [\![n+m]\!]}\big) > S_{(k)} \cap E_\delta \big) + \mathbb{P}(E_\delta^c) \\
&\leqslant \mathbb{P}\Big( s\big(X_{n+i}, R_{n+i}^{c+t}, (X_j)_{j \in [\![n+m]\!]}\big) > S_{(k)}^{\texttt{true}} \Big) + \delta \leqslant \alpha - \delta + \delta = \alpha \,.
\end{aligned}
$$

To pass from the second to the third line, we have used (11) which is satisfied on the event $E_\delta$. The last inequality is obtained using that if the score function $s(x, r, (x_j)_j)$ is invariant by permutation of its last argument and the vectors $(X_i, R_i^{c+t})_{i \in [\![n+m]\!]}$ are exchangeable, then the true scores $\{S_i^{\texttt{true}}\}_{i \in [\![n]\!]}$ are exchangeable. Therefore, we can use Theorem 1 with $k = \lceil (n+1)(1 - \alpha + \delta) \rceil$.

### A.2  Restatement of Gazin et al. [2024, Theorem 2.4]

We restate here Theorem 2.4 of Gazin et al. [2024] which controls the empirical cumulative distribution function of conformal $p$-values.

**Theorem A.1.** *Let $n, m \geqslant 2$, $V_1, \ldots, V_{n+m}$ be exchangeable real random variables having no ties almost surely. Let us, for $1 \leqslant i \leqslant n$, define the conformal $p$-values:*

$$p_i = \frac{1}{m+1} \left( 1 + \sum_{j=1}^m \mathbf{1}\{V_i \geqslant V_{n+j}\} \right) \tag{12}$$

*and $\widehat{F}_n(t) := n^{-1} \sum_{i=1}^n \mathbf{1}\{p_i \leqslant t\}$ be their empirical cumulative distribution function. Then, for $\beta \in (0, 1)$:*

$$\mathbb{P}\left[ \sup_{t \in [0,1]} \left| \widehat{F}_n(t) - I_m(t) \right| > \sqrt{\frac{\log(1/\beta) + \log\left(1 + \sqrt{2\pi\tau_{n,m}}\right)}{2\tau_{n,m}}} \right] \leqslant 2\beta \,, \tag{13}$$

*where $\tau_{n,m} = \frac{nm}{n+m}$ and $I_m(t) = \lfloor (m+1)t \rfloor / (m+1)$.*

*Proof.* The concentration inequality is obtained from Gazin et al. [2024, Theorem 2.4]. The lower deviation was enounced in its proof (Equation (38)). We apply the theorem with $r = 1$ and upper bound $\tau_{n,m}/\sqrt{n+m}$ by $\sqrt{\tau_{n,m}}/2$:

$$
\begin{aligned}
\Psi(1) = 1 \wedge \left( \frac{\log(1/\beta) + \log(1 + \sqrt{2\pi} \frac{2\tau_{n,m}}{\sqrt{n+m}})}{2\tau_{n,m}} \right) &\leqslant \frac{\log(1/\beta) + \log(1 + \sqrt{2\pi\tau_{n,m}})}{2\tau_{n,m}} \\
&\leqslant \frac{\log(2\sqrt{2\pi\tau_{n,m}}/\beta)}{2\tau_{n,m}} \,.
\end{aligned}
$$

The first term is the deviation obtained by Gazin et al. [2024]. The first upper bound is the one appearing in equation (13). The second upper bound is used to get Proposition 3.8  $\qquad\square$

## A.3 Proof of Proposition 3.8

On the event $\left\{\forall i \in [\![n]\!] : R_i^{c+t} \in \left[R_i^-, R_i^+\right]\right\}$, we have:

$$S_i^{\text{true}} := s\left(X_i, R_i^{c+t}, (X_k)_{k \in [\![n+m]\!]}\right) \leqslant \max_{r \in [R_i^-, R_i^+]} s\left(X_i, r, (X_k)_{k \in [\![n+m]\!]}\right) := S_i .$$

Hence, as earlier the quantiles of the "true" scores $S_i^{\text{true}}$ are upper bounded by the quantiles of the proxy scores $\{S_i\}_{i \in [\![n]\!]}$, i.e., for all $k \in [\![n]\!]$:

$$S_{(k)}^{\text{true}} \leqslant S_{(k)} . \tag{14}$$

Then, on this event:

$$\sum_{i=1}^m \mathbf{1}\left\{R_{n+i}^{c+t} \notin \widehat{\mathcal{C}}_k(X_{n+i})\right\} = \sum_{i=1}^m \mathbf{1}\left\{S_{n+i}^{\text{true}} > S_{(k)}\right\} \leqslant \sum_{i=1}^m \mathbf{1}\left\{S_{n+i}^{\text{true}} > S_{(k)}^{\text{true}}\right\} . \tag{15}$$

If the true scores have no ties, we can directly apply Theorem A.1 to obtain a control of the FCP. The following technical details allow to control it in the case where the score function is discrete, for instance when $s = s^{\mathbf{RA}}$. Let us introduce

$$\Gamma := \min_{\substack{1 \leqslant i,j \leqslant n+m \\ S_i^{\text{true}} \neq S_j^{\text{true}}}} \left|S_i^{\text{true}} - S_j^{\text{true}}\right|$$

and an independent family of random variables $\gamma_i$ for $i \in [\![n+m]\!]$ (also independent of $(S_i^{\text{true}})_{i \in [\![n+m]\!]}$) following a uniform distribution on $(-\Gamma/2, \Gamma/2)$. Define $\widetilde{S}_i := S_i^{\text{true}} + \gamma_i$ for $i \in [\![n+m]\!]$. Observe that $\Gamma$ depends on the true scores, but is invariant by permutation of these scores. Hence, the "perturbed" scores $\widetilde{S}_i$ are exchangeable and have (almost surely) no ties [2]. Moreover for $i \in [\![m]\!]$:

$$\left\{S_{n+i}^{\text{true}} > S_{(k)}^{\text{true}}\right\} = \left\{\sum_{j=1}^n \mathbf{1}\left\{S_{n+i}^{\text{true}} > S_j^{\text{true}}\right\} \geqslant k\right\}$$

$$\subseteq \left\{\sum_{j=1}^n \mathbf{1}\left\{S_{n+i}^{\text{true}} + \gamma_{n+i} > S_j^{\text{true}} + \gamma_j\right\} \geqslant k\right\}$$

$$= \left\{\sum_{j=1}^n \mathbf{1}\left\{\widetilde{S}_{n+i} > \widetilde{S}_j\right\} \geqslant k\right\} = \left\{\widetilde{S}_{n+i} > \widetilde{S}_{(k)}\right\} .$$

The main argument of these inequalities is that by construction of the random variables $\gamma_i$, for all $i, j$, the event $\left\{S_i^{\text{true}} > S_j^{\text{true}}\right\}$ is included in $\left\{\widetilde{S}_i > \widetilde{S}_j\right\}$.

Then we can upper bound the FCP using (15):

$$\frac{1}{m} \sum_{i=1}^m \mathbf{1}\left\{R_{n+i}^{c+t} \notin \widehat{\mathcal{C}}_k(X_{n+i})\right\} \leqslant \frac{1}{m} \sum_{i=1}^m \mathbf{1}\left\{\widetilde{S}_{n+i} > \widetilde{S}_{(k)}\right\} . \tag{16}$$

We want to apply Theorem A.1 to the exchangeable random variables $\widetilde{S}$, which have no ties. Let us express the FCP in term of conformal $p$-values. We introduce for $i \in [\![m]\!]$

$$\widetilde{p}_i := \frac{1}{n+1}\left(1 + \sum_{j=1}^n \mathbf{1}\left\{\widetilde{S}_{n+i} \geqslant \widetilde{S}_j\right\}\right)$$

and for $t \in (0, 1)$

$$\widetilde{F}_m(t) := \frac{1}{m} \sum_{i=1}^m \mathbf{1}\{\widetilde{p}_i \leqslant t\} .$$

---

[2] $\Gamma$ is not defined if all scores are equal. However, in this case the FCP is null and then upper bounded by $\alpha$.

Then for $i \in [\![m]\!]$, the following events are equal:

$$\left\{\widetilde{S}_{n+i} > \widetilde{S}_{(k)}\right\} = \left\{\sum_{j=1}^{n} \mathbf{1}\left\{\widetilde{S}_{n+i} > \widetilde{S}_j\right\} \geqslant k\right\} = \left\{\sum_{j=1}^{n} \mathbf{1}\left\{\widetilde{S}_{n+i} \geqslant \widetilde{S}_j\right\} \geqslant k\right\}$$

$$= \left\{\sum_{j=1}^{n} \mathbf{1}\left\{\widetilde{S}_{n+i} \geqslant \widetilde{S}_j\right\} > k-1\right\} = \left\{\widetilde{p}_i > k/(n+1)\right\},$$

where we have used that the scores $\widetilde{S}$ have no ties. Injecting these equalities into (16), we get:

$$\frac{1}{m}\sum_{i=1}^{m} \mathbf{1}\left\{R_{n+i}^{c+t} \notin \widehat{\mathcal{C}}_k(X_{n+i})\right\} \leqslant 1 - \widetilde{F}_m\left(k/(n+1)\right).$$

Then, with probability $1 - (\beta + \delta)$, using (13) (inverting $n$ and $m$), we get:

$$\frac{1}{m}\sum_{i=1}^{m} \mathbf{1}\left\{R_{n+i}^{c+t} \notin \widehat{\mathcal{C}}_k(X_{n+i})\right\} \leqslant 1 - I_n\left(k/(n+1)\right) + \lambda_{n,m} = 1 - \frac{k}{n+1} + \lambda_{n,m} \leqslant \alpha + \lambda_{n,m},$$

where $\lambda_{n,m} = \sqrt{\dfrac{\log(C\sqrt{\tau_{n,m}}/\beta)}{\tau_{n,m}}}$. $\qquad\square$

## A.4   Proof of Proposition 4.1

Let us apply Theorem A.1 to $V_i = Y_i$ for $1 \leqslant i \leqslant n+m$. Then the $p$-value $p_i$-s are affine transformations of the ranks of the calibration point in the test sample:

$$p_i = \frac{1}{m+1}\left(1 + R_i^t\right),$$

where we recall that $R_i^t := \mathrm{R}\left(Y_i, \{Y_{n+j}\}_{j\in[\![m]\!]}\right)$. Let us then rewrite the empirical cumulative distribution function $\widehat{F}_n$ in function of $R_i^t$, for $t \in (0,1)$:

$$\widehat{F}_n(t) = \frac{1}{n}\sum_{i=1}^{n} \mathbf{1}\left\{R_i^t \leqslant (m+1)t - 1\right\}.$$

Let us now remark that for some $u \in \mathbb{R}_+$, it is equivalent for a calibration sample point $j$ to have its rank in the test set smaller than $u$ than to have more points than its rank in the calibration with a rank smaller than $u$:

$$\forall j \in [\![n]\!]: \qquad R_j^t \leqslant u \iff \sum_{i=1}^{n} \mathbf{1}\left\{R_i^t \leqslant u\right\} \geqslant R_j^c \iff n\widehat{F}_n\left((u+1)/(m+1)\right) \geqslant R_j^c.$$

We point out that the first equivalence is satisfied as the order of the items in the calibration set is conserved when considering their insertion ranks in the test set: for all $1 \leqslant i,j \leqslant n$, $R_i^c \leqslant R_j^c \Rightarrow R_i^t \leqslant R_j^t$. Thus, for $j \in [\![n]\!]$, if $R_j^t \leqslant u$, the $R_j^c$ calibration items $i \in [\![n]\!]$ satisfying $R_i^c \leqslant R_j^c$ also satisfy $R_i^t \leqslant u$, showing the direct implication ($\Rightarrow$) in the last display. Conversely, if $k$ is the number of calibration items $i \in [\![n]\!]$ satisfying $R_i^t \leqslant u$, they are necessarily exactly the items corresponding to the $k$ smallest calibration ranks. Otherwise, there would be a calibration item $j \in [\![n]\!]$ with $R_j^c = k' > k$ and $R_j^t \leqslant u$, implying (by the previous token) that $\sum_{i=1}^{n} \mathbf{1}\{R_i^t \leqslant u\} \geqslant k' > k$, a contradiction. Thus any calibration item with $R_j^c \leqslant k$ satisfies $R_j^t \leqslant u$.

Thus, with probability $1 - 2\delta$, thanks to (13), for all $t \in (0,1)$:

$$\left|\widehat{F}_n(t) - I_m(t)\right| \leqslant \lambda, \quad \text{where} \quad \lambda = \sqrt{\frac{\log(1/\delta) + \log\left(1 + \sqrt{2\pi\tau_{n,m}}\right)}{2\tau_{n,m}}}.$$

Assume this event is satisfied. Then in particular, for $u = (m+1)\left(R_j^c/n + \lambda\right)$:

$$n\widehat{F}_n\left((u+1)/(m+1)\right) = n\widehat{F}_n\left(R_j^c/n + \lambda + 1/(m+1)\right)$$

$$\geqslant nI_m\left(R_j^c/n + \lambda + 1/(m+1)\right) - n\lambda$$

$$= n\lfloor(m+1)\left(R_j^c/n + \lambda\right) + 1\rfloor/(m+1) - n\lambda$$

$$\geqslant n\left(R_j^c/n + \lambda\right) - n\lambda \geqslant R_j^c.$$

Thus, $R_j^t \leqslant (m+1)(R_j^c/n + \lambda)$ for $1 \leqslant j \leqslant n$. Let us now lower bound these ranks. Similarly, we have the equivalence for $u \in \mathbb{R}$ and $1 \leqslant j \leqslant n$:

$$R_j^t > u \iff \sum_{i=1}^n \mathbf{1}\{R_i^t \leqslant u\} \leqslant R_j^c \iff n\widehat{F}_n\big((u+1)/(m+1)\big) \leqslant R_j^c.$$

With $u = (m+1)(R_j^c/n - \lambda)$, we have $n\widehat{F}_n\big((u+1)/(m+1)\big) \leqslant R_j^c$ and then $R_j^t \geqslant (m+1)(R_j^c/n - \lambda)$ for $1 \leqslant j \leqslant n$ which concludes the proof. $\qquad\square$

## A.5 Proof of Proposition 4.2

The proof is based on the concentration bound of Naaman [2021] recalled below which is a recent multi-dimensional version of the Dvoretzky–Kiefer–Wolfowitz (DKW) inequality [Dvoretzky et al., 1956].

**Theorem A.2** (Naaman, 2021). *Let $X^{(1)}, \dots X^{(K)}$ i.i.d. random vectors in $\mathbb{R}^n$. Then for $t \geqslant 0$,*

$$\mathbb{P}\left[\sup_{\theta \in \mathbb{R}^n} \left|F(\theta) - \widehat{F}_K(\theta)\right| \geqslant t\right] \leqslant n(K+1)e^{-2Kt^2}, \tag{17}$$

*where $F(\theta) = \mathbb{P}\left[X_i^{(1)} \leqslant \theta_i, \forall i\right]$ and $\widehat{F}_K$ is the empirical cdf defined by:*

$$\widehat{F}_K(\theta) = \frac{1}{K}\sum_{k=1}^K \mathbf{1}\left\{X_i^{(k)} \leqslant \theta_i, \forall i\right\}.$$

**Proof of Proposition 4.2.** For $\theta \in \mathbb{R}^n$, let us denote

$$F(\theta) = \mathbb{P}\left[\forall i \in [\![n]\!] : R_{(i)}^{c+t} \leqslant \theta_i\right]$$

and $\widehat{F}_K$, the empirical cdf:

$$\widehat{F}_K(\theta) = \frac{1}{K}\sum_{k=1}^K \mathbf{1}\left\{R_i^{(k)} \leqslant \theta_i, \forall i \in [\![n]\!]\right\}.$$

Let us now remark that:

$$\mathbb{P}\left[\forall i \in [\![n]\!] : R_i^{c+t} \in \left[R_{R_i^c}^-, R_{R_i^c}^+\right]\right] = \mathbb{P}\left[\forall i \in [\![n]\!] : R_{(i)}^{c+t} \in \left[R_i^-, R_i^+\right]\right]$$
$$= \mathbb{E}\left[F(R^+) - F(R^-)\right].$$

The ordered ranking vector $R_{(i)}^{c+t}$ follows indeed the same distribution than the drawn vectors $R_i^{(k)}$. Then:

$$F(R^+) - F(R^-) = \left(F(R^+) - \widehat{F}_K(R^+)\right) - \left(F(R^-) - \widehat{F}_K(R^-)\right) + \widehat{F}_K(R^+) - \widehat{F}_K(R^-)$$
$$\geqslant -2\sup_{\theta \in \mathbb{R}^n}\left|F(\theta) - \widehat{F}_K(\theta)\right| + (1 - \delta),$$

as by assumption on the envelope, $\widehat{F}_K(R^+) - \widehat{F}_K(R^-) \geqslant 1 - \delta$ almost surely. Then after taking the expectation, we get:

$$\mathbb{P}\left[\forall i \in [\![n]\!] : R_i^{c+t} \in \left[R_{R_i^c}^-, R_{R_i^c}^+\right]\right] \geqslant 1 - \delta - 2\mathbb{E}\left[\sup_\theta\left|F(\theta) - \widehat{F}_K(\theta)\right|\right].$$

It remains to upper bound that expectation. Let us denote $Z = \sup_\theta\left|F(\theta) - \widehat{F}_K(\theta)\right|$, let $t \geqslant 0$,

$$\mathbb{E}[Z] = \mathbb{E}[Z\mathbf{1}_{Z\leqslant t} + Z\mathbf{1}_{Z>t}] \leqslant t + \mathbb{P}[Z > t] \leqslant t + n(K+1)e^{-2Kt^2}.$$

We have used that $Z$ is bounded by 1 and Theorem A.2. We conclude by choosing $t = \sqrt{\frac{\log nK}{K}}$. $\quad\square$

# B Alternative targets

We detail in this section the guarantees we get on the alternative target sets evoked in Section 3.3.

**Rank among the test points.** A user might be interested in the ranking of the test points among themselves, rather than within the entire dataset. Such a set can be constructed directly from the previous sets, as presented in following Corollary B.1.

**Corollary B.1.** *For $i \in [\![m]\!]$, let*

$$a_{n+i} = \min \widehat{\mathcal{C}}_k(X_{n+i}), \quad \text{and} \quad b_{n+i} = \max \widehat{\mathcal{C}}_k(X_{n+i}), \tag{18}$$

$$N_{n+i}^+ = |\{j \in [\![n]\!], R_j^+ \leqslant a_{n+i}\}| \quad \text{and} \quad N_{n+i}^- = |\{j \in [\![n]\!], R_j^- \leqslant b_{n+i}\}|. \tag{19}$$

*Then $\widehat{\mathcal{C}}_k^t(X_{n+i}) = [a_{n+i} - N_{n+i}^-, b_{n+i} - N_{n+i}^+]$ is marginally valid, $\mathbb{P}[R_{n+i}^t \in \widehat{\mathcal{C}}_k^t(X_{n+i})] \geqslant 1 - \alpha$ and has a controlled FCP, with probability $1 - \beta - \delta$:*

$$\frac{1}{m} \sum_{i=1}^m \mathbf{1}\left\{R_{n+i}^t \notin \widehat{\mathcal{C}}_k^t(X_{n+i})\right\} \leqslant \alpha + \lambda_{n,m}, \tag{20}$$

*with $\lambda_{n,m}$ defined in Proposition 3.8 .*

*Proof.* As in the proof of Theorem 2.1, we assume that the event $E_\delta = \{\forall j \in [\![n]\!] : R_j^{c+t} \in [R_j^-, R_j^+]\}$ holds. Then if $R_{n+i}^{c+t} \in \widehat{\mathcal{C}}_k(X_{n+i}) \subset [a_{n+i}, b_{n+i}]$ it implies that $R_{n+i}^t \in [a_{n+i} - R_{n+i}^c, b_{n+i} - R_{n+i}^c]$. Thus, we can upper and lower bound $R_{n+i}^c$ by just noticing that $R_{n+i}^c = |\{j \in [\![n]\!], R_j^{c+t} \leqslant R_{n+i}^{c+t}\}|$. It follows:

$$N_{n+i}^+ \leqslant |\{j \in [\![n]\!], R_j^{c+t} \leqslant a_{n+i}\}| \leqslant R_{n+i}^c \leqslant |\{j \in [\![n]\!], R_j^{c+t} \leqslant b_{n+i}\}| = N_{n+i}^-,$$

as $R_j^{c+t} \in [R_j^-, R_j^+]$. It gives the marginal coverage as, then:

$$[a_{n+i} - R_{n+i}^c, b_{n+i} - R_{n+i}^c] \subset [a_{n+i} - N_{n+i}^-, b_{n+i} - N_{n+i}^+].$$

The control of the FCP is obtained in the same manner using Proposition 3.8. $\qquad\square$

**Top-K items.** It is also possible to target the top-K items by selecting all items whose prediction set includes at least one rank smaller than or equal to K. Corollary B.2 exhibits that this set contains in average $K - \alpha m$ of the top K items. This can be sufficient if we target a "top" proportion of the items. This strategy is used in Section E.1 to identify top-1 candidates in the Yummy Food dataset.

**Corollary B.2.** *For $K \in [\![m]\!]$, let $\widehat{\mathcal{C}}^{topK} = \{X_{n+i} : \exists \ell \leqslant K, \ell \in \widehat{\mathcal{C}}_k(X_{n+i})\}$. Then*

$$\mathbb{E}\left[\left|\widehat{\mathcal{C}}^{topK} \cap \mathcal{C}^{topK}\right|\right] \geqslant K - \alpha m, \tag{21}$$

*where $\mathcal{C}^{topK} = \{X_{n+i} : R_{n+i}^{c+t} \leqslant K\}$ is the set of the true top K items.*

*Proof.* Let us just link the expectation size to the coverage:

$$
\begin{aligned}
|(\widehat{\mathcal{C}}_k^{topK})^c \cap \mathcal{C}_k^{topK}| &= \left|\left\{i \in [\![m]\!] : R_{n+i}^{c+t} \leqslant K, \text{ and } \forall \ell \in [\![K]\!], \ell \notin \widehat{\mathcal{C}}_k(X_{n+i})\right\}\right| \\
&\leqslant \left|\left\{i \in [\![m]\!] : R_{n+i}^{c+t} \leqslant K, \text{ and } R_{n+i}^{c+t} \notin \widehat{\mathcal{C}}_k(X_{n+i})\right\}\right| \\
&\leqslant \left|\left\{i \in [\![m]\!] : R_{n+i}^{c+t} \notin \widehat{\mathcal{C}}_k(X_{n+i})\right\}\right| \\
&= \sum_{i=1}^m \mathbf{1}\left\{R_{n+i}^{c+t} \notin \widehat{\mathcal{C}}_k(X_{n+i})\right\}.
\end{aligned}
$$

By taking the expectation, we get that $\mathbb{E}\left[|(\widehat{\mathcal{C}}_k^{topK})^c \cap \mathcal{C}_k^{topK}|\right] \leqslant \alpha m$ and then $\mathbb{E}\left[|\widehat{\mathcal{C}}_k^{topK} \cap \mathcal{C}_k^{topK}|\right] \geqslant K - \alpha m$. $\qquad\square$

# C  Details on the procedure

## C.1  Quantile envelope

Algorithm 2 explicitly details the construction of the quantile envelope, estimated by Monte Carlo with a sample of size $K$.

---

**Algorithm 2** Quantile envelope procedure

---

1: **Input:** $n, m, K$, and $\delta \in (0,1)$, $(R_i^c)_{i \in [\![n]\!]}$
2: **for** $k = 1, \ldots, K$ **do**
3:     $\tilde{R}^{(k)} \leftarrow$ Output of **Algorithm 1**
4: **end for**
5: $\widehat{\gamma} \leftarrow \max \gamma$
    such that: $\sum_k \mathbf{1}\left\{ \exists i : \tilde{R}_i^{(k)} \notin \left[ \widehat{\mathsf{Q}}_\gamma\left( (\tilde{R}_i^{(\ell)})_\ell \right), \widehat{\mathsf{Q}}_{1-\gamma}\left( (\tilde{R}_i^{(\ell)})_\ell \right) \right] \right\} \leqslant K\delta$
6: **for** $j = 1, \ldots, n$ **do**
7:     $\widehat{R}_j^- \leftarrow \widehat{\mathsf{Q}}_{\widehat{\gamma}}\left( (\tilde{R}_j^{(\ell)})_{\ell \in [\![K]\!]} \right)$
8:     $\widehat{R}_j^+ \leftarrow \widehat{\mathsf{Q}}_{1-\widehat{\gamma}}\left( (\tilde{R}_j^{(\ell)})_{\ell \in [\![K]\!]} \right)$
9: **end for**
10: **Output:** $\widehat{R}_{R_i^c}^-, \widehat{R}_{R_i^c}^+$ for $i \in [\![n]\!]$.

---

In this algorithm, for $x \in \mathbb{R}^K$ and $u \in (0,1)$, $\widehat{Q}_u(x)$ denotes the empirical quantile of order $u$ of the sample $x$.

## C.2  FCP control

In this section, we give details on the procedure to control the FCP at level $\bar{\alpha} \in (0,1)$. First, let us remark that FCP $:= m^{-1} \sum_{i=1}^m \mathbf{1}\{p_i \leqslant t\}$, where the $p_i$ are given in Eq. (12). In other words, the FCP is the proportion of conformal p-values that are lower than $t$. Furthermore, we have that [Gazin et al., 2024]:

$$\{\text{FCP} \leqslant \bar{\alpha}\} = \left\{ m^{-1} \sum_{i=1}^m \mathbf{1}\{p_i \leqslant t\} \leqslant \bar{\alpha} \right\} = \left\{ \sum_{i=1}^m \mathbf{1}\{p_i \leqslant t\} \leqslant \lfloor m \cdot \bar{\alpha} \rfloor \right\} = \left\{ p_{(\lfloor m \cdot \bar{\alpha} \rfloor + 1)} > t \right\},$$

where $p_{(a)}$ denotes the $a$-th smallest value among $(p_1, \ldots, p_m)$. We see that controlling the FCP at level $\bar{\alpha}$ is equivalent to control that the ordered p-value $p_{(\lfloor m \cdot \bar{\alpha} \rfloor + 1)}$ is strictly larger than $t$. Hence, for $\bar{\beta} > 0$, if we denote by $t_{\bar{\beta}}$ the quantile of level $\bar{\beta}$ of the distribution of $p_{(\lfloor m \cdot \bar{\alpha} \rfloor + 1)}$, then $\mathbb{P}(\text{FCP} \leqslant \bar{\alpha}) = \mathbb{P}\left( p_{(\lfloor m \cdot \bar{\alpha} \rfloor + 1)} > t_{\bar{\beta}} \right) \geqslant 1 - \bar{\beta}$.

It remains to find $t_{\bar{\beta}}$. As proven by Gazin et al. [2024, Proposition 2.1], if the scores $V_1, \ldots, V_{n+m}$ are exchangeable random variables having no ties almost surely, the vector of conformal p-values $(p_1, \ldots, p_m)$ follows a known universal distribution. Hence, it is possible to simulate samples following the same distribution as $p_{(\lfloor m \cdot \bar{\alpha} \rfloor + 1)}$ by first simulating the vector $(p_1, \ldots, p_m)$ and then by tacking the $(\lfloor m \cdot \bar{\alpha} \rfloor + 1)$ smallest value. Then, we compute the empirical quantile of order $\bar{\beta}$ to find $t_{\bar{\beta}}$. The full procedure is given in Algorithm 3.

# D  Additional results on synthetic data

We present in this section additional experiments on the synthetic data of Section 5.1 and on another synthetic data set especially constructed to highlight the adaptivity of the score $s^{\mathbf{VA}}$.

## D.1  Additional results for synthetic data of Section 5.1

We complete the results of Section 5.1 when instead of using RankNet we use:

**Algorithm 3** Control of the FCP

1: **Input:** $\bar{\alpha}, \bar{\beta}, n, m$ and $K$
2: $\tilde{P} \leftarrow$ vector of size $K$
3: **for** $k = 1, \ldots, K$ **do**
4:      Draw $n + m$ uniform random variable $U_i$ on $[0, 1]$
5:      **for** $i = 1, \ldots, m$ **do**
6:          $\tilde{p}_i \leftarrow \frac{1}{m+1}\left(1 + \sum_{j=1}^{m} \mathbf{1}\{U_{j+n} \geqslant U_i\}\right)$
7:      **end for**
8:      $\tilde{P}_k \leftarrow \widehat{Q}_{\lfloor m \cdot \bar{\alpha}\rfloor + 1}(\tilde{p}_1, \ldots, \tilde{p}_m)$
9: **end for**
10: $\widehat{t}_{\bar{\beta}} \leftarrow \widehat{Q}_{\bar{\beta}}(\tilde{P})$
11: **Output:** $\widehat{t}_{\bar{\beta}}$

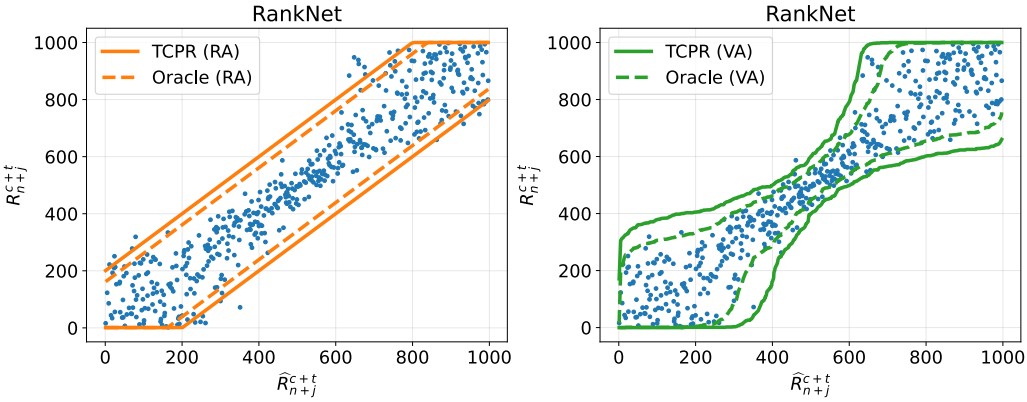

Figure 5: Synthetic data: True ranks $R_{n+j}^{c+t}$ in function of their predicted rank $\widehat{R}_{n+j}^{c+t}$ by RankNet and their prediction sets with scores $s^{\mathbf{RA}}$ and $s^{\mathbf{VA}}$ for $n = m = 500$.

- LambdaMART with 100 trees of 5 leaves trained on $n_{tr} = 300$ data points. Performances, in term of FCP and relative length, for different values of $n \in \{100, 500, 2500\}$ and $m \in \{100, 500, 2500\}$ are presented in Figure 9.

- RankSVM trained with $n_{tr} = 300$. Performances for different values of $n \in \{100, 500, 2500\}$ and $m \in \{100, 500, 2500\}$ are presented in Figure 10.

- BRE applied with $1\%$ of all the possible comparisons. Performances for different values of $n \in \{100, 500, 2500\}$ and $m \in \{100, 500, 2500\}$ are presented in Figure 11. Note that BRE is of different nature than the three other algorithms as it does not learn a score from their features but ranks items using a small number of pairwise comparisons. The ranking errors are due to this limited number of comparisons.

We also provide performances when using RankNet for $n \in \{100, 500, 2500\}$ and $m \in \{100, 2500\}$ in Figure 12. Note that in all the synthetic experiments, we use RankNet with a ReLU Neural Network (NN) of 5 hidden layers of size 10. This NN is trained using `Pytorch` [Paszke, 2019]. Figure 5 displays an example of the prediction sets we construct with TCPR and with the oracle method. All the numerical results are postponed to Section E.3 for clarity of presentation.

**Results:** As already observed in the main paper, our method, TCPR, always control the false coverage proportion at level $\alpha = 0.1$. Furthermore, at any given size $m$, as the size $n$ of the calibration set increases, our methods return better sets with FCP closer to $0.1$ and increasingly smaller sizes. In general, we need more calibration points than the number of test points to be able to construct prediction sets that are not too large, which is quite intuitive. For $n$ large relatively to $m$, we reach the performance of the oracle. For LambdaMART and RankSVM, the score function $s^{\mathbf{RA}}$ gives in general better prediction sets than those constructed with $s^{\mathbf{VA}}$. However, this difference is less notable for BRE. Nevertheless, it should be noted that sets with $s^{\mathbf{RA}}$ are less adaptive than the

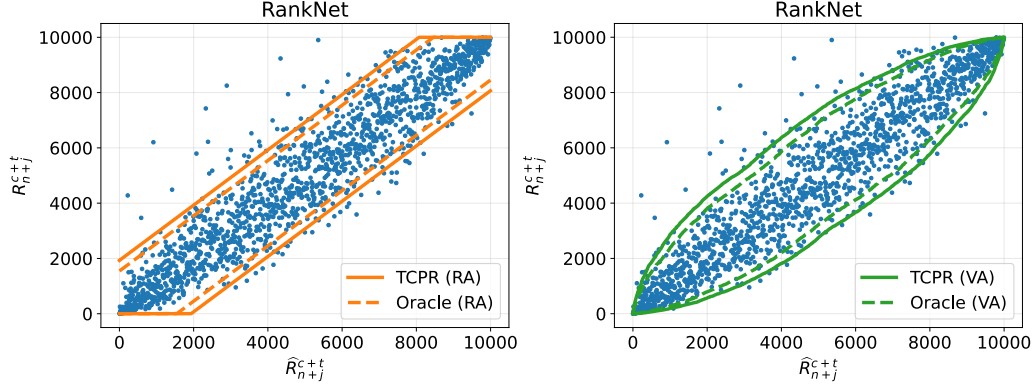

Figure 6: Yummly-10k: True ranks $R_{n+j}^{c+t}$ in function of their predicted rank $\widehat{R}_{n+j}^{c+t}$ by RankNet and their prediction sets with scores $s^{\textbf{RA}}$ and $s^{\textbf{VA}}$.

ones with $s^{\textbf{VA}}$ as explain in Section 5.1. Overall, the difference of performance between the score functions depends on the distribution of the data and the algorithm.

# E    Additional results on real data

In this section, we present additional information and results on the two real data sets considered in the main paper.

## E.1    Yummly-`10k`

**Hyper-parameters:** For all the experiments on this data set, we use RankNet with a ReLU NN of $5$ hidden layers of size $10$. This NN is trained using `PyTorch` [Paszke, 2019].

**Additional results:** Figure 6 illustrates an example of the prediction sets returned by our method and by the oracle. As already seen from Table 1, the differences between TCPR and the oracle are relatively small. This can be explained by the large size of the data set which makes the impact of the envelopes negligible. Notice that from these predictions sets, we can infer who the potential candidates of rank 1 would be. Figure 7 shows these candidates when the prediction sets are constructed using TCPR or the oracle method with $s^{\textbf{VA}}$. The candidates are all the images containing the rank 1 in their prediction set, i.e. we display $\left\{ \text{dishes X} : 1 \in \widehat{\mathcal{C}}(X) \right\}$. As the prediction set of the image of true rank 1 contains 1 with high probability, this set will contain this image with high probability. The image highlighted in red is the randomly chosen reference $x^*$ which is, by definition, the true rank 1 dish (see Section 5.2). Firstly, we observe that the reference $x^*$ is included among these top-1 candidates and that the other dishes are closely related to it. Secondly, as expected, the number of candidates is greater for TCPR (14 candidates) than for Oracle (10 candidates) but not by much. Thirdly, remark that, due to the use of $s^{\textbf{VA}}$, the prediction sets are narrower for the lower ranks, so the number of candidates remains relatively low. For this specific task, the score $s^{\textbf{VA}}$ is particularly more adapted than $s^{\textbf{RA}}$.

## E.2    Anime recommendation LTR data set

**Data set:** In details, this data set consists in the three following lists[3]:

1. A list of $16681$ movies with the following associated features: name, genres, is_tv, year, is_adult, above_five_star_users, above_five_star_ratings, above_five_star_ratio.

2. A list of $15163$ users with the following associated characteristics: review_count, user_feature avg_score, user_feature score_stddev, user_feature above_five_star_count, user_feature, above_five_star_ratio.

[3]`https://www.kaggle.com/datasets/ransakaravihara/anime-recommendation-ltr-dataset`

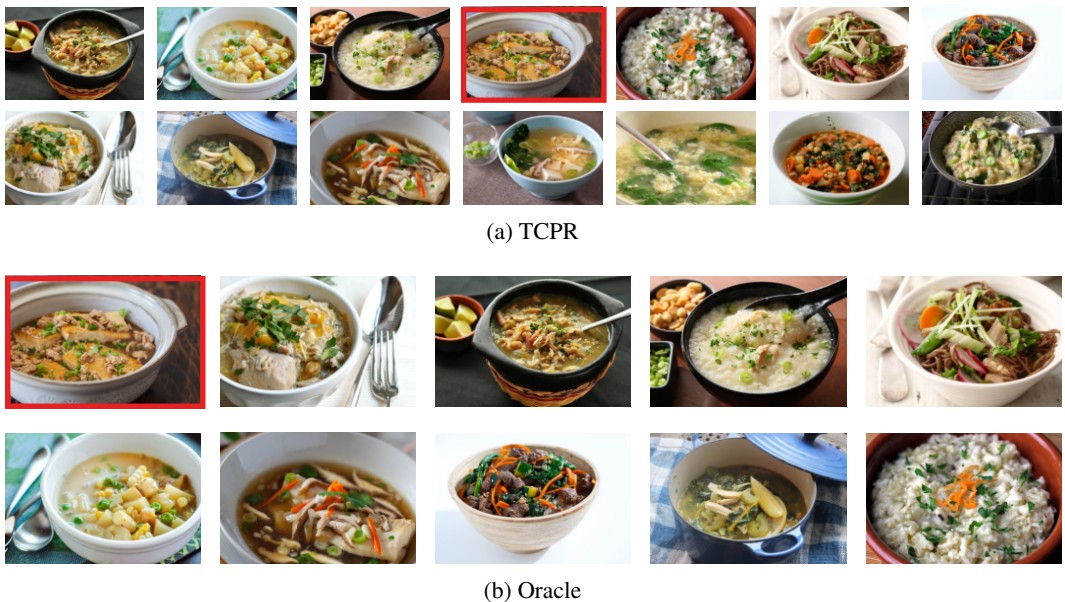

(a) TCPR

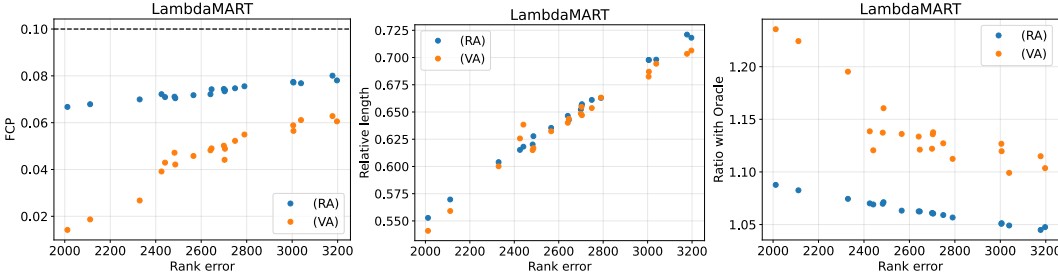

(b) Oracle

Figure 7: Candidates for the rank $1$ dish when using TCPR or the oracle methods with $s^{\mathbf{VA}}$. The reference $x^*$ (which is the true rank $1$ dish) is highlighted in red.

Figure 8: Results on the anime recommendation LTR data set. Each point corresponds to a model LambdaMART(tree,leaves) trees $\in \{50, 100, 200, 300\}$ and leaves $\in \{5, 10, 15, 20\}$. The FCP, relative length, and ratio with the oracle are displayed for each score in function of the average rank error of each model.

3. A list of ratings for $10^6$ tuples (movie, user), with values ranging from $0$ to $10$.

**Ranking task and calibration:** We use LambdaMART from the LightGBM implementation[4] for this LTR problem. The model assigns a score to each (user, movie) tuple and has been trained on a subset of $15000$ users. The reference model has $400$ trees and $20$ leaves, the smaller ones have trees $= \{50, 100, 200, 300\}$ and leaves $= \{5, 10, 15, 20\}$. For a particular user, we then suppose only having access to the scores predicted by the large model for $n = 2000$ anime, and evaluate the others $m = 16681 - n$ scores with the smaller models.

**Results:** Figure 8 displays the different metrics with respect to the rank error of all the LambdaMART models. The uncertainty across different instances is well captured by our procedure, as illustrated in the second row of the plots: the size of the prediction intervals increases with the ranking algorithm's errors. For all methods, the FCP is well controlled and remains below the threshold of $0.1$. Once again, we observe that the envelope effect is more pronounced for high-performing algorithms: the ratio with the oracle decreases as the ranking error increases. Nevertheless, the ratios remain close to $1$ for all methods, suggesting that the overall effect of the envelope is minor.

---

[4] https://github.com/microsoft/LightGBM

## E.3 Additional numerical results

This section contains additional experiments for the synthetic dataset (different tuples $(n, m)$).

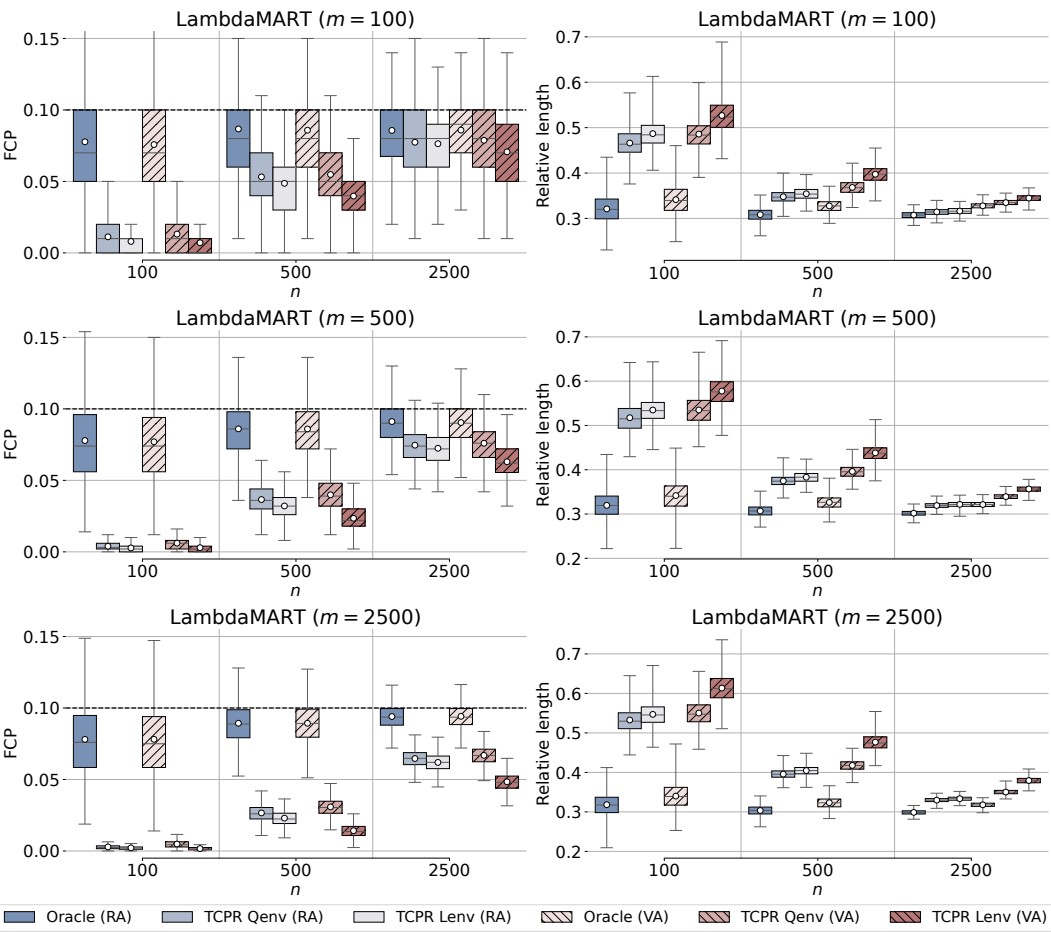

Figure 9: Synthetic data: FCP and relative lengths obtained for LambdaMART with the (**RA**) and (**VA**) score, for the quantile (*Qenv*) and linear (*Lenv*) envelopes when $m = \{100, 500, 2500\}$ and $n \in \{100, 500, 2500\}$. White circles represent the means.

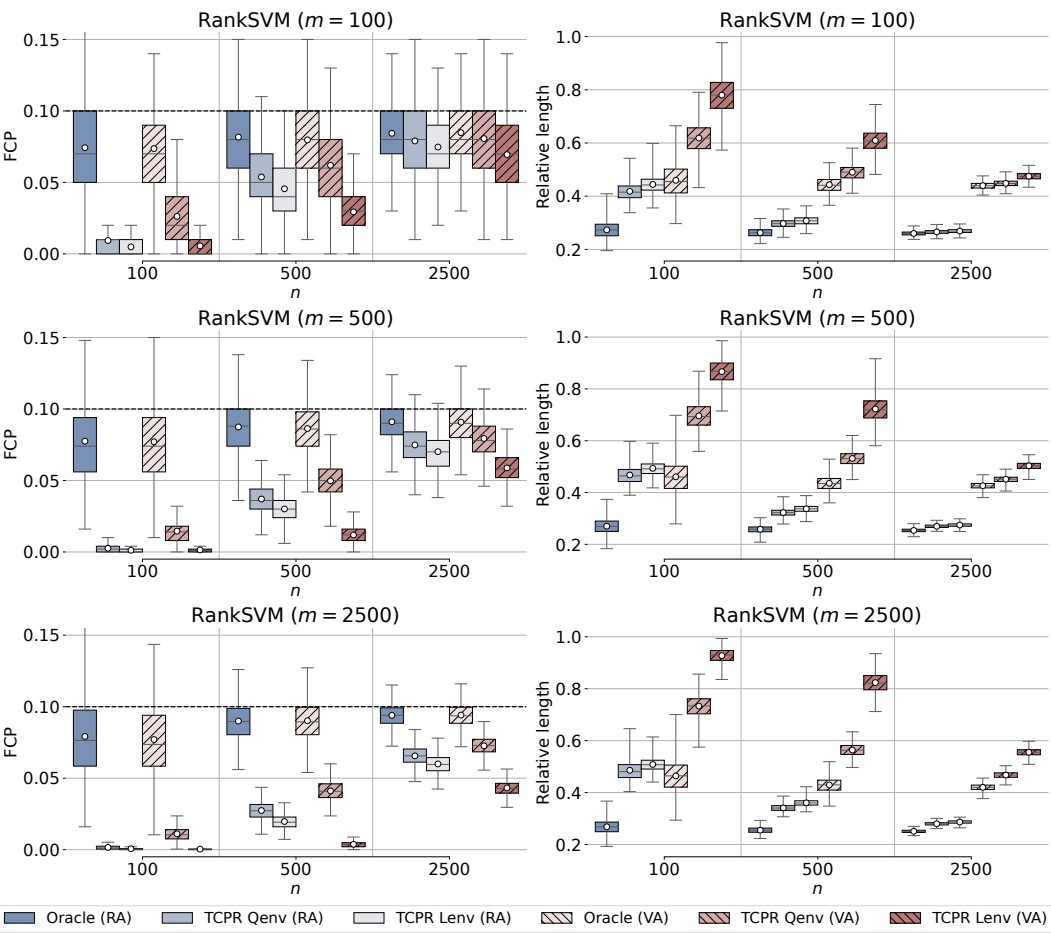

Figure 10: Synthetic data: FCP and relative lengths obtained for RankSVM with the (**RA**) and (**VA**) score, for the quantile (*Qenv*) and linear (*Lenv*) envelopes when $m = \{100, 500, 2500\}$ and $n \in \{100, 500, 2500\}$. White circles represent the means.

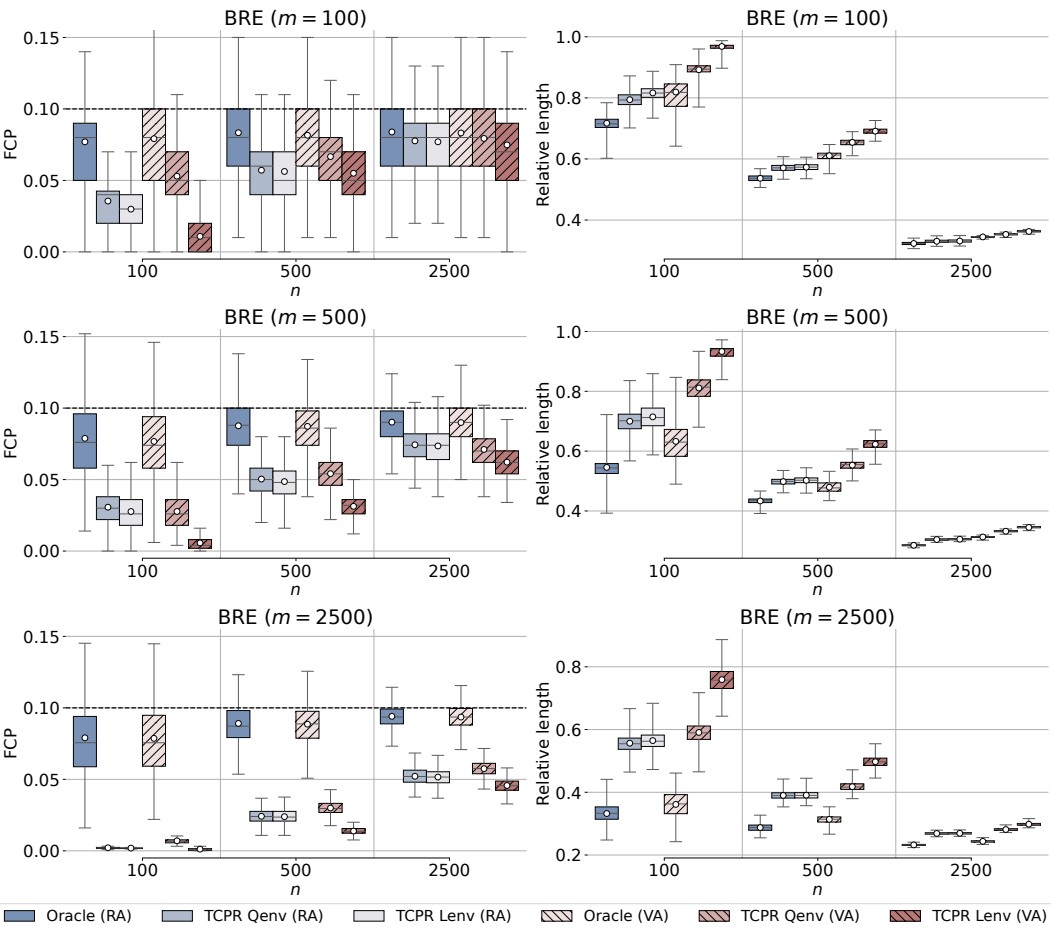

Figure 11: Synthetic data: FCP and relative lengths obtained for BRE with the (**RA**) and (**VA**) score, for the quantile (*Qenv*) and linear (*Lenv*) envelopes when $m = \{100, 500, 2500\}$ and $n \in \{100, 500, 2500\}$. White circles represent the means.

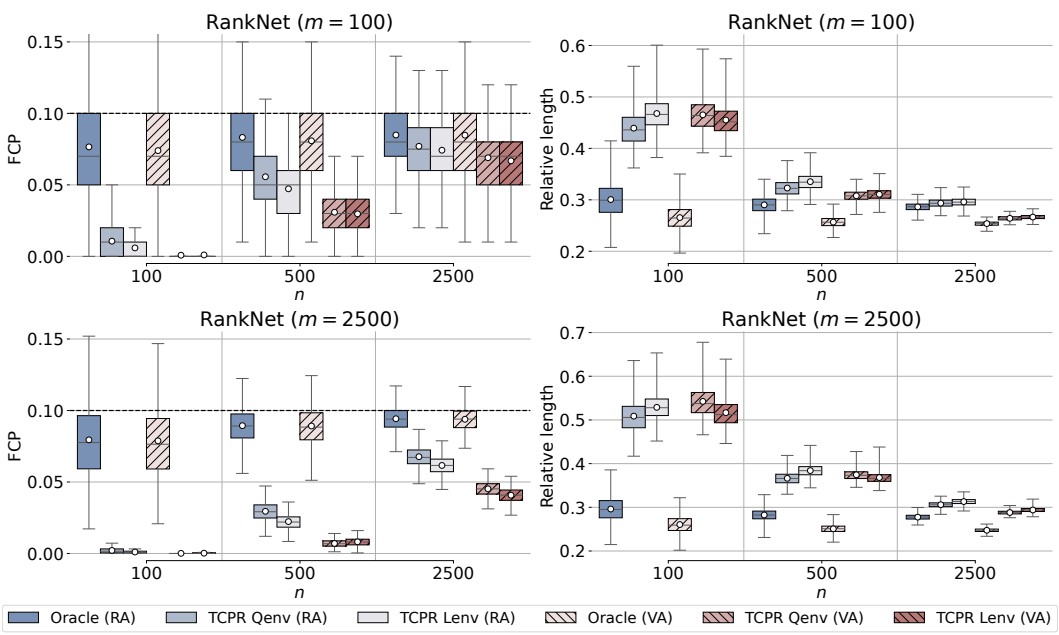

Figure 12: Synthetic data: FCP and relative lengths obtained for RankNet with the (**RA**) and (**VA**) score, for the quantile (*Qenv*) and linear (*Lenv*) envelopes when $m = \{100, 2500\}$ and $n \in \{100, 500, 2500\}$. White circles represent the means.

