# OpenReview forum: "Transductive Conformal Inference for Full Ranking"
_NeurIPS.cc/2025/Conference — NeurIPS 2025 poster_

### Official Review · Reviewer_MSYV · 2025-06-14

**Clarity:** 2
**Significance:** 2
**Originality:** 2
**Rating:** 3
**Confidence:** 4

**Summary:**

This work provides a methodology for constructing prediction sets for the ranks among unobserved outcome variables.

**Questions:**

1.  I would recommend that the authors further elaborate on the problem formulation and motivation. At this point, the target problem feels somewhat artificial. Specifically:

1-1. I'm not entirely sure whether constructing prediction sets for the ranks would be practically useful. Inference on the ranks can certainly be a meaningful problem, but I'm not convinced that it necessarily needs to take the form of prediction sets. In the experiments, there is an example with $m = 2500$. What can a practitioner realistically learn from the 2500 prediction sets for the ranks, and in what situations would one prefer this form of inference?

If it is inference on the outcome $Y_{n+j}$'s, prediction set sounds natural, but if we are interested in the relative order between the outcomes, I feel that a more useful target would be to select the individuals with the largest values among them (with some guarantee), or to test if the ranks are correct, etc.

1-2. Also, the target of inference is set as the "rank among all the $n + m$ points" rather than the "rank among the $m$ test points," which seems less useful to me. I believe the test set is typically the "set of targets of interest" in inference problems, but in this work, the target essentially includes the calibration data as well---whose within-calibration set ordering is known---so it is not very clear why this is set as the target of inference.

1-3. I cannot think of a realistic setting where we observe only the relative orders of $Y$, but not the actual values, in the calibration set. In the examples mentioned in the paper---“new players entering the game that we want to rank among the previous players to organize matches between players with similar skills” or “movies that we want to rank among those already known by a particular user for which we have his feedback”---it does not seem that the data would actually take the form of (feature, relative rank). Rather, it would more naturally be in the form of (feature, outcome)---e.g, (feature, skill evaluation score) or (feature, feedback score).

2. The proposed methodology is said to be a conformal-type method that requires only exchangeability, but the resulting formula seems to be based on concentration-type bounds. Is this an approximation of some explicit finite-sample formula (as in conformal prediction) introduced to address potential computational challenges, or is there no such underlying finite-sample expression? (I know "finite-sample formula" is not an accurate expression, but hopefully the meaning is clear.)

In the experiments, it is also stated that the proposed method performs well for large $n$, such as 2500, but is not very informative when $n = 100$, so it does not feel to me that the method is indeed a kind of conformal inference.

3. Minor comments
- line 139 $(Y_i)_i$, line 154 $(X_j)_j$ : seem like typos?

- In line 302, it says that Figure 2 shows the results under $m = 2500$, but the caption of Figure 2 states $m = 500$.

- There are some issues with the referencing of theorems, propositions, and assumptions.

**Ethical Concerns:**

["NO or VERY MINOR ethics concerns only"]

**Final Justification:**

My major concerns were not resolved---the problem setting seems quite artificial and do not seem much useful to me.

**Limitations:**

-

**Paper Formatting Concerns:**

-

**Quality:**

2

**Strengths And Weaknesses:**

The paper provides rigorous theoretical arguments.

However, the writing requires significant improvement—in terms of sentence structure, notation, and overall organization.
The motivation is unclear, and the problem formulation appears somewhat artificial, seemingly designed to fit an existing approach. This makes the proposed methodology appear not very useful in practice, despite its theoretical appeal.

---

> ### Author Rebuttal · Authors · 2025-07-30
>
> We thank the reviewer for their constructive questions regarding the conformal nature of our method. Below, we clarify why our approach can indeed be considered conformal. We also provide additional references and explanations to support the practical relevance of both the setting and the type of data we consider, points that the reviewer has questioned, and on which we respectfully disagree.
>
> **1) Problem formulation and motivation**
>
> **1-1 "constructing prediction sets for the ranks would be practically useful":**
>
> We do not defend that constructing all the intervals is always useful. One can simply use our method to obtain a prediction set for the rank of a single item of interest, for which our method provides the first conformal methodology, to the best of our knowledge. By adjusting the level, we have demonstrated that the method also provides control over the False Coverage Proportion (FCP) if the user wishes to apply it to all items, but this is an additional benefit.
>
> A major advantage of controlling the FCP is that the numerous prediction sets can be combined. For example, one can derive a set containing the top-$k$ items with high probability by including all items whose prediction set contains at least one rank smaller than or equal to $k$ (see Section D.1 and Figure 7). Using the control of the FCP of Proposition 3.8, this set contains, with high probability, at least approximately $k-\alpha m$ items among the top-$k$ ($\alpha m$ being the worst number of miscovered points). We will provide a more detailed explanation of this application.
>
> **1-2 "the target of inference is set as the "rank among all the points" rather than the "rank among the test points," which seems less useful to me.":**
>
> We do not entirely agree with your remark. While it is true that, in some situations, only the rank of the $m$ new items is of interest, it is not always the case. We thus consider the ranking problem among $n+m$ items in the paper as it arises in certain applications (see below) and, moreover, is more general than ranking solely the $m$ new items.
>
> Here are two applications considered in the paper where we want to rank the $n+m$ items:
> - The arrival of new items to be ranked among already known items. This arrival may follow a temporal dynamic with successive entries; see, for example, [1]. We simulated this type of scenario in Section 5.2 using the Yummy Food dataset.
> - The combination of a powerful but costly ranking algorithm with a less performant but more efficient one. The powerful algorithm may correspond to a heavy architecture requiring significant computational or energy cost, or alternatively, to manual comparisons performed by humans. We considered this type of setting in Section 5.3 with the Anime recommendation dataset.
>
> In addition, note that we can already derive sets for the rank among the $m$ test points from our prediction sets on the $n+m$ points. We only need to shrink all the sets by the number of calibration points of smaller rank. Let $C_{k}(X_{n+j}) = [a_{n+j},b_{n+j}]$, $N_{n+j}^{-} = |\lbrace i: 1\leq i \leq n, R_{i}^{+} < a_{n+j} \rbrace| $ and $N_{n+j}^{+} = | \lbrace i: 1\leq i \leq n, R_i^+ \leq b_{n+j}  \rbrace |$, then $[a_{n+j} -N_{n+j}^-,  b_{n+j} -N_{n+j}^+]$ are prediction sets for the rank among the $m$ test points. They satisfy a marginal coverage, and the FCP satisfies (9), using that we have a uniform control of the rank of the calibration points (Eq. 6). We will add this discussion in the final version of the paper.
>
> **1-3 "I cannot think of a realistic setting where we observe only the relative orders of $Y$":**
>
> There are many contexts in which only pairwise comparisons between items are observed, as the target notion $Y$ is difficult, or even impossible, to measure. Common examples include taste, skill, emotion, and more. In fact, the literature of *object ranking* focuses on constructing a global ranking from such comparisons (see, for example, [2] for a survey).
>
> The Yummy Food dataset, for instance, is constructed from triplet-wise comparisons of dishes precisely because a direct value $Y$ of the notion of taste is not measurable. As other examples, we can mention the dataset IMDB-WIKI-SbS [3], which is a dataset constructed from pairwise comparisons of pictures of people to determine which people are older and capture the subjective human opinion. In psychology, [4] aims to measure how people are perceived by others to highlight potential biases in the population. This dataset is constructed from pairwise comparisons of people to determine which one seems the most friendly (a subjective value difficult to measure directly).
>
> Finally, often, these comparisons are used to compute a score, such as the Elo rating, which is derived from sequences of pairwise matches. Nevertheless, the original observations remain comparisons between items, and in this last example, players.
>
> **2) "the resulting formula seems to be based on concentration-type bounds"**
>
> In our paper, we use a concentration result derived from [5] (recalled in Theorem A.1). This result is distribution-free and assumes only exchangeability. In fact, we agree that the deviation bound $\lambda_{n,m}$ present in the bound of Proposition 3.8 could be replaced by the exact quantile of the distribution-free statistics $\|\| F_{n} - I_{m}\|\|_{\infty}$ (see Eq. 13). However, since we are not aware of an exact expression for such a quantile, we use a concentration bound for our theoretical results. In practice, we use either an estimation of this quantity (the linear envelope), or the quantile envelope, which adaptively controls the distance between $F_n$ and $I_m$. Both methods perform better than the theoretical one, as illustrated in Figure 1, and have theoretical coverage guarantees (Proposition 4.2). Finally, while we effectively use a finite sample for estimating these in practice, we do not provide a "conformal analysis" as it is not straightforward to link the quantile envelope to a score function.
>
> **-- "it does not feel to me that the method is indeed a kind of conformal inference."**
>
> We respectfully disagree with the reviewer's comment. While it is true that in CP, the number of calibration points does not significantly influence coverage, it does greatly affect the informativeness of the prediction sets, i.e. their length. Indeed, the length of a CP set often depends heavily on the number of calibration samples. As a pathological but illustrative example, when the number of calibration points is smaller than $\simeq\alpha^{-1}$, the resulting prediction set becomes overly conservative as it must contain all possible values.
>
> The influence of the number of calibration points on the lengths of the sets is observable in our experiments (in the different boxplots): the size of the sets decreases as the number of calibration points $n$ increases. With more calibration data, our method can better capture the error of the ranking algorithm, leading to tighter and thus more informative prediction sets.
>
> Overall, we want to reaffirm here that our method remains a conformal one in the rigorous sense: it guarantees a coverage, a False Coverage Proportion (FCP) in our experiments, that exceeds the target level $1-\alpha$, regardless of the values of $n$ (or $m$) and the distribution of the data i.e. our method is distribution-free.
>
> **3) Typos**
>
> We thank the reviewer for identifying some typos in the paper. We have made the necessary corrections.
>
>
> We hope we have addressed your concerns and are happy to discuss further if needed.
>
> [1] Learning to Rank for Multi-Step Ahead Time-Series Forecasting, Duan and Kashina, (2021).
>
> [2] A Survey and Empirical Comparison of Object
> Ranking Methods, Kamishima et al., (2010).
>
> [3] IMDB-WIKI-SbS: An Evaluation Dataset for Crowdsourced Pairwise Comparisons, Pavlichenko and Ustalov, (2021).
>
> [4] Person Perception Biases Exposed: Revisiting the First Impressions Dataset, Junior et al. (2020).
>
> [5] Transductive conformal inference with adaptive scores, Gazin et al. (2024).

---

> > ### Comment · Reviewer_MSYV · 2025-08-03
> >
> > Thank you for the detailed response. Below are the questions I think need more clarification:
> >
> > In responses 1–3, the explanation "because a direct value of the notion of taste is not measurable": If there is no well-defined $Y$ and only comparisons are made, I believe there is no guarantee that the set of comparisons leads to a consistent full ordering. So it seems to me that in most practical settings, there has to be some $Y$ that is simply unobserved, and only their relative ordering is observed---for the theory in this work to be applicable. In the experiment section, it is stated that a ranking algorithm is used, but I am not sure whether it aligns with the theory, which is based on actual ranks from exchangeable data points.
> >
> > In response 2, the statement "While it is true that in CP, the number of calibration points does not significantly influence coverage, it does greatly affect the informativeness of the prediction sets, i.e., their length": this seems to apply only in settings with extremely small sample sizes. For example, if $\alpha = 0.1$, I don't think the difference between $n = 100$ and $n = 200$ would be substantial. I asked this question because the experiments were conducted with much larger sample sizes.

---

> > > ### Author Response · Authors · 2025-08-04
> > > **Answer to comment of Reviewer MSYV**
> > >
> > > Thank you for your additional comments and for engaging the discussion.
> > >
> > > **1-** We fully agree with the comment that "there has to be some $Y$ that is simply unobserved, and only their relative ordering is observed", which corresponds to the setting we consider. When we state that "$Y$ is not measurable", we mean that it is unobservable, although an underlying $Y$ does exist and defines a full ranking. This constitutes a modeling choice that is commonly adopted in the ranking literature.
> > >
> > > We also would like to recall that the ranking algorithms we consider learn a function that ranks items based on their *features*. This function is typically learned from a data set consisting of partial orders over items (along with their corresponding features), usually obtained through pairwise comparisons. The goal is to learn a function that mimics the underlying true value $Y$, which is only observed through these comparisons.
> > >
> > > More specifically, in the paper we distinguish between two situations: the case where the algorithm directly assigns a rank to each item (e.g., the BRE algorithm based on pairwise comparisons, see Section C.1), and the case where it learns a scoring function from the features to construct a ranking (e.g., RankNet, LambdaMART, or RankSVM). The uncertainty we quantify using conformal prediction (CP) differs in each case: in the first, it arises from the limited number of pairwise comparisons; in the second, it stems from the uncertainty in constructing the scoring function.
> > >
> > > In the experiments presented in the main paper, we consider the second case, where RankNet, LambdaMART, and RankSVM are trained on a separate dataset (a standard approach in split CP). We then construct prediction sets for the $n + m$ items based on the ranks predicted by the ranking algorithm and the known relative ranks of the calibration set.
> > >
> > >
> > > **2-** We agree that the influence of the number of calibration points on the size of the prediction set is not always striking. However, we think important to keep in mind that, although conformal prediction (CP) is distribution-free, it relies on quantile estimation, which in some cases may require a sufficiently large number of calibration points to achieve the necessary precision. Below is an example of such a pathological case, illustrating that the prediction set size can vary significantly even for large $n$.
> > >
> > > Consider a classification setting in which, for the correct class, the score is $0$ with probability $p = (0.9+ 10^{-9})$, and $1$ otherwise and the scores for the incorrect classes are always $1$. For $\alpha = 0.9$, the optimal threshold is then $0$, which would yield correct coverage with a prediction set of size $1$. However, the quantile of the scores will oscillate between $0$ and $1$ while $1/\sqrt{n} \gtrsim 10^{-9}$. If the threshold ends up being $1$, then the prediction set will include all the possible labels. For sufficiently large $n$ (in this case, extremely large), the prediction set stabilizes to a singleton corresponding to the correct label.
> > >
> > > In our experiments, we have considered large-scale data sets because in ranking tasks, for instance in information retrieval task, data sets can contain millions of documents.

---

> > > > ### Comment · Reviewer_MSYV · 2025-08-04
> > > >
> > > > Thank you for your additional responses.
> > > >
> > > > After the additional discussions, I remain unconvinced that the problem setting and the proposed procedure offer practical utility. Therefore, I will maintain my original score.
> > > >
> > > > Regarding Point 1, I feel the main question was not adequately addressed. Specifically, I was asking: What are some realistic scenarios where the setting in this work applies? More pointedly, how can ranks from ranking algorithms—when there is no well-defined Y---be used to run the proposed method, which was designed for actual ranks derived from some exchangeable Y?
> > > >
> > > > As for Point 2, I do not believe that such an extreme mathematical example is helpful in discussing the performance of the proposed method.

---

> > > > > ### Author Response · Authors · 2025-08-04
> > > > > **Answer to second comment of Reviewer MSYV**
> > > > >
> > > > > **1-** We effectively have misunderstood the point of the reviewer and try to clarify. On real data sets, we effectively used as true ranks the rank produced by a more complex process.
> > > > >
> > > > > For Yummly data set, the ground truth is the distance to a randomly chosen unknown item. For Anime dataset, the ranks are given by a ranking algorithm with a larger architecture. We consider these quantities as the true ranks and then, our theoretical guarantees only state that our prediction sets contain (w.h.p.) the ranks that would have been given by these processes. We proceed like this to simulate the costly process of crowdsourcing or conducting user surveys, which are commonly  used in practice to have access to a ground truth (e.g. the Yummly dataset).
> > > > >
> > > > > Regarding the theoretical guarantees, we only consider this process as the underlying truth and apply our results. To be specific, let $\tilde{\mathcal{A}}$ be this proccess, for Yummly data set it is the distance to the reference point and, for Anime data set, the rank provided by the algorithm. Then, if the data are exchangeable and $\tilde{\mathcal{A}}$ treats symetrically the calibration and the test sets (e.g. RankNet trained separatly or the distance to a reference point), the underlying truth $Y_i=\tilde{\mathcal{A}}(X_i)$ remains exchangeable and our theoretical results apply.
> > > > >
> > > > > We sincerely hope that this addresses the reviewer's questions.
> > > > >
> > > > > **2-** This is indeed a pathological example, but overconfident classifiers can unfortunately exhibit such behavior. We included it in response to the reviewer’s comment in response 2 to illustrate that the size of a conformal prediction set can be significantly affected by variations in the number of calibration points, even when this number is very large. This example supports the broader discussion about the conformity of our method and relates to point 2 of the rebuttal.

---

### Official Review · Reviewer_C52t · 2025-07-03

**Clarity:** 4
**Significance:** 3
**Originality:** 4
**Rating:** 5
**Confidence:** 3

**Summary:**

This paper studies the problem of uncertainty quantification of full ranking algorithms relying on black box predictions. In particular, they develop conformal prediction sets for the correct ranking of each element that satisfies 1) marginal coverage for each element and 2) controls the joint false coverage ratio of the sets. In their method, as they only have partial calibration data (meaning only have access to the true ranking of the some of the elements in calibration time), they propose a conformal calibration method relying on lower and upper bounds on the unseen ground truth elements. Furthermore, they propose several ways to come up with these bounds. They also numerically evaluate the performance of their method.

**Questions:**

- What is the role of ranking structure here? what i mean by that is, in normal CP, oftentimes there is not much relation between different labels. However, here, it might not make sense if you find yourself in as situation that you are uncertain whether the ranking of en element is 1 or 10, but certain that its not 2 to 9. In other words, it looks like to me that there might be some structure specific to the ranking problem that one might be able to exploit to further improve the algorithm.

- An adjacent problem to what you are considering, which might also be interesting for a range of real-world applications would be: lets imagine the same setting which you are working on. But this time, instead of asking for one prediction set for the ranking of each element, one might be interested in what is the element with rank 1? and subsequently a prediction set that quantify the uncertainty of the rank 1. That is to say, a prediction set that includes the rank 1 with high probability. Any comment whether you method gets extended to these scenarios?

**Ethical Concerns:**

["NO or VERY MINOR ethics concerns only"]

**Final Justification:**

I have read the other reviews and I am happy to keep my score of acceptance, as I think this is a novel application of CP, with some interesting theoretical challenges that the authors have overcome. That being said, I also believe further clarifications and empirical evaluations (e.g. to the works that reviewer 4soE mentioned) are needed to understand the depth of the practical value of this paper.

**Limitations:**

yes.

**Quality:**

3

**Strengths And Weaknesses:**

- This is a novel application of CP to ranking problems. In particular, due to the partial nature of calibration data in this setting, it raises some statistical challenges, which the authors take care of carefully, that might be of independent interest beyond the application of ranking.
- I like the presentation of the paper. The problem formulations are well motivated and well explained.
- The theoretical guarantees and arguments are sound to me.

---

> ### Author Rebuttal · Authors · 2025-07-30
>
> We sincerely thank the reviewer for their thoughtful and positive review. We answer her/his interesting questions below.
>
> **1) Role of ranking structure**
>
> This is an interesting point. In our method, we implicitly use the ranking structure through the choice of the score functions. The two score functions we consider can be interpreted as residual scores and provide a set of the form of an interval (see lines 188 and 191 for the definitions of the related sets). This implies that we will not select ranks far away from each other in the prediction set without selecting the ranks between them. The situation you mention is then *hopefully* not possible.
>
> One possibility to further leverage the structure of the ranking problem, which we have considered but not explored in depth, is to transform the resulting sets into sets of possible permutations. Since we provide possible ranks for each item, it is possible that some values may be incompatible, as the ranks are dependent on each other. Therefore, we can remove these incompatible values as a post-processing step, thereby reducing the number of possible rankings. However, we have considered that this method might lose readability and thus did not include it in the paper.
>
> **2) Top k prediction set**
>
> Thank you once again for this interesting question. We have mentioned this extension in the conclusion of the paper. To obtain a set containing the top-k items with high probability, one possibility is to select all items whose prediction set includes at least one rank smaller than or equal to $k$. We applied this strategy to identify the top-1 item in the Yummy Food dataset, as detailed in Section D.1. Using Proposition 3.8, this set contains at least approximately the top $k-\alpha m$ of the items. This can be sufficient if we target a "top" proportion of the items. To have specific guarantee for the top-1 item, we have different directions to explore in order to adress this question. One idea could be to use techniques from the framework of selective CP, where the selection is defined as the "top-1." Another one could be to view this problem as a distribution-shift problem. This is because the distribution of a point, given that its rank is $1$, is not the same as that of the other points. Thus, we could potentially rely on weighted conformal prediction techniques.

---

### Official Review · Reviewer_eSy5 · 2025-07-03

**Clarity:** 3
**Significance:** 3
**Originality:** 2
**Rating:** 4
**Confidence:** 3

**Summary:**

This paper addresses the challenge of quantifying the uncertainty for full ranking tasks using conformal prediction (CP) methods, specifically in scenarios where only the relative ranks of a portion of the data are known and new items need to be ranked amongst them. The authors introduce a transductive conformal prediction approach that does not assume access to full calibration rankings, instead constructing non-trivial bounds for the unknown ranks based on recent results for the distribution of conformal p-values. The work establishes theoretical guarantees (including False Coverage Proportion control), proposes both explicit and Monte Carlo envelope construction methods for bounding unknown ranks, and empirically evaluates the methods on synthetic and real datasets using several learning-to-rank algorithms.

**Questions:**

The envelope is simulated from uniform order statistics assuming exchangeability. How often do the real calibration ranks fall outside your proposed envelope on actual datasets? Have you empirically validated this high-probability inclusion assumption?

Classical CP yields sets valid conditional on observed features. Your sets are valid only under marginal rank-based assumptions. How can practitioners trust these sets in the presence of covariate shift or model miscalibration?

You compute max conformity scores over artificial rank intervals not tied to observed data or residuals. Why is this preferable to a data-driven bootstrap or nonconformity-based estimation?

**Ethical Concerns:**

["NO or VERY MINOR ethics concerns only"]

**Final Justification:**

Thank you for the detailed and thoughtful rebuttal. Your clarifications regarding the data-driven nature of the proxy scores, and the justification for the envelope construction based on the universal distribution of conditional ranks, helped address several of my earlier concerns. In particular, I now better understand how the method uses observed features and model outputs in constructing the conformity scores, and how the simulation-based envelope still retains distribution-free validity under the exchangeability assumption.

I still have some reservations, particularly regarding the method’s sensitivity to practical violations of exchangeability (e.g., when the model is trained or tuned on overlapping data), and the limited adaptivity of the envelope to model-specific or instance-level uncertainty. These concerns may affect the method's reliability in real-world deployments. That said, I appreciate the authors’ candid acknowledgment of these limitations and the clear positioning of their contribution within the standard CP framework.

Given these clarifications, I am raising my score.

**Limitations:**

Yes

**Paper Formatting Concerns:**

No major formatting issues. The paper generally adheres to NeurIPS 2025 formatting guidelines.

**Quality:**

2

**Strengths And Weaknesses:**

**Strengths**:

The paper addresses a timely and practically important problem: quantifying the uncertainty of ranking outputs when full calibration data is unavailable, a situation common in real-world applications.

The main methodological contribution is clearly described: adapting split/transductive conformal prediction to ranking problems with partial calibration, using recent theoretical results to construct valid prediction sets. This is a nontrivial extension of CP.

The authors derive marginal validity and false coverage proportion (FCP) bounds under a carefully constructed framework. The methodology is built on top of CP theory, including bounds for joint distributions of p-values.

The experiments are thorough, spanning synthetic data and real-world ranking tasks.

**Weaknesses**:

The methodology relies critically on exchangeability of the pair $(X_i, R_i^{c+t})$, and more importantly, on the assumption that tight high-probability bounds $[R_i^−, R_i^+]$ can be obtained for the calibration items’ true ranks. These assumptions are rarely satisfied or testable in real applications, particularly when the ranking algorithm is trained on data or is adaptive.

Constructing rank envelopes via Monte Carlo simulations of order statistics is not grounded in observed data. The resulting bounds on calibration ranks are not conditional on observed $X_i$ or model predictions, and are not adaptively learned from the actual calibration set. As a result, the prediction sets are not meaningful for the observed data realization, and any formal guarantee is only marginal over hypothetical permutations, not conditionally valid in practice.

In classic CP, conformity scores are grounded in residuals between predictions and ground truth. Here, the proxy scores are computed over artificial intervals—again not informed by the true data realization. Consequently, the prediction sets are not robust to model misspecification or data shift.

---

> ### Author Rebuttal · Authors · 2025-07-30
>
> We deeply thank the reviewer for her/his feedback and questions. It appears that there may have been a misunderstanding regarding the use of our data in the proposed method. We would like to draw the reviewer's attention to the fact that the data are indeed used to calibrate the ranking algorithm’s error, as explained in our point a) below.
>
>
> **a) Weaknesses: A possible misunderstanding of the proxy scores**
>
> We want to begin with a comment on a particular point regarding our paper that seems to have led to an important misunderstanding. Indeed, it seems that the reviewer thinks that we do not use the data to compute the proxy scores as she/he said in the review that:
>
> *"The proxy scores are computed over artificial intervals—again not informed by the true data realization"*.
>
> We would like to clarify here that this is not the case. The scores are not computed from artificial data but using the ground-truth pairs $(X, R^c)$ from the calibration set. Indeed, $\hat{R}^{c+t}$ is constructed via the ranking algorithm and therefore uses the features $X$ while the computation of the scores uses $\hat{R}^{c+t}$ and $R^c$. The part on the envelope is only necessary to bound the unobserved variable $R^t$ (the rank of the item among the test set). As $R^{c+t} = R^c+ R^t$, we then get a bound on the whole rank and then, are able to evaluate the error of the ranking algorithm on the calibration data, as done in standard conformal prediction methods. The variable $R^t$ is linked to the concept of conformal p-values, which have a known distribution. This allows us to approximate its distribution via Monte Carlo method for computational efficiency. However, importantly, since everything is distribution-free, we can theoretically control this approximation to still provide finite sample guarantees (see Proposition 4.2). We believe this is an important distinction to make.
>
> **b) Weaknesses: Exchangeability of $(X, R)$ and tight bounds for $R_{-}$ and $R_{+}$.**
>
> We would like to address the comment that:
>
> *"these assumptions are rarely satisfied or testable in real applications."*
>
> It is true that in Theorem 3.5, we assume the availability of high probability bounds for $R^{c+t}$ (Eq. 6)  in order to have a general statement. While we present this as an assumption, in Section 4, we demonstrate how to construct such bounds, both theoretically and empirically. In fact, as shown in Proposition 4.1, the ability to construct such bounds is a consequence of the exchangeability of the pairs $(X, R)$ and (Eq. 6) is therefore satisfied as soon as the data are exchangeable. We have chosen this presentation to allow the user to incorporate alternative bounds that they may find more appropriate, for example, by using external information.
> Note also that we may have misunderstood your point on the difficulty of having exchangeability. As you mention, for adaptive algorithms this property is indeed not necessary verify. However, this is inherently satisfied for most ranking algorithms (for example the ones in the experiments) when the data are i.i.d., and the calibration data are not used for the training. Therefore, this assumption seems to us to be reasonable and is quite standard in conformal prediction framework.
>
> **1.1) Uniform order statistics and calibration ranks**
>
> First, recall that we use the envelope to approximate the unobserved true rank of the calibration $R^{c+t}$ because only the ranks $R^c$ are observed. We are able to compute the envelope using uniform order statistics (Algorithm 2) because the distribution of $R^{c+t}$ knowing $R^c$ is universal (see discussion l.233). In other words, the distribution of $R^{c+t}$ knowing $R^c$ does not depend on the distribution of the initial data. We then rely on this distribution-free property in our method.
>
> **1.2) How often do the real calibration ranks fall outside the envelope on actual datasets?**
>
> Theoretically, under the exchangeability assumption, we know that the envelope will contain with high probability all the ranks of the calibration points (Proposition 4.1). We will add numerical results illustrating this property in the final version of the paper.
> We would like to point out that we have compared our method to an oracle that uses the true ranks of the calibration data to compute the scores. Since our method approaches the oracle very closely as $n$ increases (Figure 2, Table 1), this also empirically demonstrates that the envelopes we use are reasonable.
>
>
>
> **2.1) Validity of the set only under marginal rank-based assumption**
>
> We refer the reviewer to the beginning of our rebuttal (Answer a)) regarding this point and recall that our prediction sets depend on $X$ as we use the groundtruth pairs $(X, R^c)$ to compute the scores. Furthermore, as shown in the experiments (boxplots and Figures 4, 5, and 6) our prediction sets are close to the one returned by the oracle that does not require the envelope. This suggests that our bound is tight. We also would like to recall that having distribution-free coverage guarantees, conditionally on the features, is impossible for non-trivial sets [1]. Our coverage guarantees are then in high probability over the randomness of the calibration and the test sets, as standard CP methods.
>
> **2.2)  How can practitioners trust these sets in the presence of covariate shift or model miscalibration?**
>
> This is an interesting remark. We agree that our method is not robust to a covariate shift. This is a common problem for CP methods, and different solutions exist [2, 3, 4]. We have considered this question outside the scope of this work, but this is obviously a very interesting line of research for future work. One possibility could be to use weighted quantiles instead of standard ones, as in the framework of weighted CP. Another strategy could be to rely on recent asymptotic results on weighted conformal p-values, for instance [5].
> Regarding model miscalibration, we are not sure we understand what you mean by that. Could you elaborate on it?
>
> **3)  Why is this preferable to a data-driven bootstrap or nonconformity-based estimation?**
>
> To begin, we would like to reaffirm here that we do indeed use the observed data $(X, R^c)$ to compute our prediction sets. Indeed, through the non-conformity scores, we incorporate the information provided by the variable $X$ and $R^c$ (see also Answer a) above). Hence, our method is indeed a data-driven method.
>
> Regarding the two points of the question:
>
> - The main advantages of our CP method over a bootstrap strategy are that we provide strict finite sample marginal guarantees, meaning our sets are guaranteed to contain the true rank $(1-\alpha) \cdot 100\%$ of the time, regardless of the number of calibration points and the distribution of the data. This is not the case for confidence intervals constructed with bootstrap technique, which generally rely on asymptotic theory.
>
> - For nonconformity-based estimation, we are not sure whether we have well understood the remark. If you mean that we could simply compute the scores as in standard CP, this strategy is simply not feasible in our ranking problem: In standard CP, computing the nonconformity scores requires the ground truth of the calibration points, here their rank $R^{c+t}$ among the $n+m$ points. In our ranking setting, this information is not fully available as we only observe the relative rank $R^c$ of the calibration. As $R^{c+t} = R^c+ R^t$, we use the notion of envelope to approximate the ground truth.
>
> We hope our answers have clarified our approach and addressed the reviewer’s concerns.
>
> [1] Conditional validity of inductive conformal predictors, Vovk, V. (2012).
>
> [2] Conformal prediction under covariate shift. Tibshirani, R. J., Foygel Barber, R., Candes, E., Ramdas, A. (2019).
>
> [3] Conformal prediction beyond exchangeability. Barber, R. F., Candes, E. J., Ramdas, A., Tibshirani, R. J. (2023).
>
> [4] Efficient conformal prediction under data heterogeneity. Plassier, V. et. al. (2024).
>
> [5] Asymptotics for conformal inference. Gazin, U. (2024).

---

> > ### Comment · Reviewer_eSy5 · 2025-08-07
> >
> > Thank you for the detailed and thoughtful rebuttal. Your clarifications regarding the data-driven nature of the proxy scores, and the justification for the envelope construction based on the universal distribution of conditional ranks, helped address several of my earlier concerns. In particular, I now better understand how the method uses observed features and model outputs in constructing the conformity scores, and how the simulation-based envelope still retains distribution-free validity under the exchangeability assumption.
> >
> > I still have some reservations, particularly regarding the method’s sensitivity to practical violations of exchangeability (e.g., when the model is trained or tuned on overlapping data), and the limited adaptivity of the envelope to model-specific or instance-level uncertainty. These concerns may affect the method's reliability in real-world deployments. That said, I appreciate the authors’ candid acknowledgment of these limitations and the clear positioning of their contribution within the standard CP framework.
> >
> > Given these clarifications, I am raising my score.

---

### Official Review · Reviewer_4soE · 2025-07-04

**Clarity:** 2
**Significance:** 2
**Originality:** 2
**Rating:** 4
**Confidence:** 4

**Summary:**

This paper introduces a method for constructing conformal prediction intervals for ranks under two settings: the RA setting and the VA setting. It employs a proxy conformal score function to compute nonconformity scores and demonstrates that the method achieves an upper-bounded false coverage rate (FCR) at level α. Various numerical experiments further support the effectiveness of the approach.

**Questions:**

Motivation and Practical Utility:
It would be helpful if the authors could more clearly articulate the motivation for applying conformal prediction methods to ranks. For instance, if the subjects’ covariates and outcomes are exchangeable—and therefore their ranks as well—wouldn't the resulting prediction intervals for ranks be quite wide? A discussion on this potential limitation would add clarity. In addition, it would be informative to evaluate the length of the predicted intervals alongside the false coverage rate (FCR), as long intervals may reduce practical utility despite valid coverage.

Interval Width and Tightness of Bounds:
The proposed method constructs prediction intervals for ranks using both test and calibration data by estimating upper and lower bounds. However, since the true ranks of new (test) observations are inherently uncertain, there is a concern that the resulting intervals may be overly conservative. Could the authors provide further empirical or theoretical insights into whether these bounds tend to be tight or overly loose in practice?

Related Work on Ranking Inference:
There is a growing body of literature on ranking inference that also focuses on constructing confidence intervals for ranks. The authors may consider acknowledging or comparing their work with some recent contributions in this area, such as:

Lagrangian Inference for Ranking Problems, Liu, Y., Fang, X., and Lu, J. (2023)

Uncertainty Quantification in the Bradley–Terry–Luce Model, Gao, G., Shen, Y., and Zhang, A. (2023)

Ranking Inference from Top Choices in Multiway Comparisons, Fan, J., Lou, Z., Wang, W., and Yu, M. (2025)

Including a discussion of these works would help better position the current contribution within the existing literature.

**Ethical Concerns:**

["NO or VERY MINOR ethics concerns only"]

**Final Justification:**

I have read the other reviews and the authors' rebuttal. I am comfortable keeping my original score, but I am also open to raising it to a 4. My only remaining concern is the practical applicability of the method, as the prediction intervals may be quite wide.

**Limitations:**

See the questions

**Quality:**

2

**Strengths And Weaknesses:**

Strength: This paper is the first to propose a conformal prediction method for ranks, which is a novel and valuable contribution to the literature.

Weakness: One concern is that the resulting prediction intervals may be too wide in practice, potentially limiting their utility—for example, as illustrated in Figure 3.

---

> ### Author Rebuttal · Authors · 2025-07-30
>
> We sincerely thank the reviewer for her/his constructive feedback and insightful questions. Below, we address each point in detail. We also appreciate the suggestions regarding the additional references, we will discussed them in the final version (see answer $3)$ below).
>
> **a) Weakness: Size too wide**
>
> We agree that the prediction sets returned by our method can sometimes be large. However, it is important to note that this is also true for the prediction sets returned by the oracle method (i.e. the best possible method). Therefore, we can conclude that the large size of our sets is mostly due to the inherent complexity of the problem and not because of our methodology. Moreover, as demonstrated empirically in the paper, i) when the ranking task is easier, for example for central ranks of Figure 5, our sets are then really small when we use the score (VA), ii) the size of our prediction sets tends to converge to those of the oracle (e.g. see metrics of Table 1, or visual comparisons of Figures 4, 5 or 6).
>
> Concerning your remark on Figure 3, although we agree that the sets appear wide, we also see that this width is necessary to ensure that 90% of the blue dots fall within our prediction sets. Furthermore, in Figures 4, 5 and 6, we visually compare our sets to those of the oracle and see that the difference is small, further supporting our previous findings. Notice that this conclusion was also present in the synthetic experiments where boxplots related to the metric "relative length" for our method and the oracle are very close (at least when $n$ is large).
>
> **1) Motivation and Practical Utility**
>
> **1.1) Why applying CP methods to ranks?**
>
> CP for ranking algorithms provides valid prediction sets for the ranks of new items. Being able to provide prediction sets for rank is an important topic in the ranking literature as this is particularly useful in decision-making where understanding the confidence of predictions is crucial (see e.g. the papers you mention). One key advantage of CP over other existing techniques is its distribution-free property. This means that, if we have an algorithm that ranks items based on their characteristics, we can determine, without making any assumptions about the underlying distribution other than exchangeability, whether the algorithm is confident in its ranking or not. This is a very important property in practice as, in general, it is hard to verify any assumptions on the data. Notice that, as discussed below in answer 3), this property is not satisfied in the papers mentioned in your review, as they rely on the Bradley-Terry-Luce (BTL) model.
>
> **1.2) Exchangeability of subjects’ covariates and outcomes**
>
> As in standard CP, to prove that our prediction sets are valid, we rely on the assumption that non-conformity scores are exchangeable. More specifically, as you mentioned, we assume that the data are exchangeable, which implies that their ranks are also exchangeable. However, this assumption does not necessarily lead to excessively large sets. Indeed, through the non-conformity scores, we incorporate the information provided by the variable $X$. Your concern about the size of the sets would be valid if the scores were constructed using only the variables $R$ and without using $X$. In standard CP, this would be equivalent to constructing scores using only $Y$ and not from the absolute residuals $\lvert Y−f(X) \rvert$. In our article, we follow a similar idea to that of absolute residuals but adapted to the ranking problem and thus incorporate the information of the $X$. Our sets are of the form (predicted rank depending on the features $X$) $\pm$ (uncertainty computed by CP). If the rank predicted from the features is high, then the prediction set will include correspondingly high ranks.
>
> **1.3) Length of the predicted intervals alongside the FCP**
>
> We are not entirely certain about what you mean by "plotting the length alongside the FCP" but if you are suggesting varying $\alpha$ and comparing the size of the sets with the FCP, we will add a figure to address this. Above is a table with the average relative length of the predicted sets and the FCP obtained on the Anime data set for some values of $\alpha$:
>
> | Level $\alpha$  | 0.1  | 0.2  | 0.3  | 0.4  |
> | --------------- | ---- | ---- | ---- | ---- |
> | FCP             | 0.05 | 0.16 | 0.25 | 0.34 |
> | Relative Length | 0.72 | 0.60 | 0.51 | 0.44 |
>
> You can observe that the length decreases as long as the target confidence level $\alpha$ is increasing.
>
> Overall, we fully agree with your remark that it is important to look at the size of the sets which is why we give them in all our experiments (e.g. Figures 3, 5, and 6) and to validate the proxy scores, we have compared ourselves to an oracle CP method, which has access to the ground truch and does not use proxy scores. We would also like to point out that we have also looked how the size of the predicted sets evolves as a function of the performance of the ranking algorithm in Figure 8. This is different from what you suggest but we think that these experiments also show that our method captures effectively the uncertainty of the ranking algorithm (also see Figure 4, score VA as discussed in Answer a).
>
> **2) Evidences on the tightness of the bounds**
>
> Thank you for this interesting question. Recall that we provide theoretical support in Proposition 4.1, where we prove that our bounds converge to zero at an order of $O(m/\sqrt{\tau_{n, m}})$. This deviation is tight, as it relies on the concentration properties of the conformal p-value, whose asymptotic behavior has been thoroughly analyzed in [4]; see Theorem 3.1 of this work. Furthermore, as mentioned above in Answer (a), our experiments empirically demonstrate that the sets returned by our methods are very similar to those of the oracle method; at least when $n$ is large enough (see, for example, the right panel of Figure 2). This indicates that the non-conformity proxy scores, that we construct using our bounds (Eq. 7), are very similar to those obtained without bounds (i.e., the oracle scores). Therefore, this suggests that our bounds are tight enough to not perturb the conformal procedure. As suggested by the reviewer, we conducted a quick additional experiment comparing the quantiles of the proxy scores and the oracle scores obtained in the Anime data set (Section 5.3). The results in the following table show that their values are indeed similar, confirming our claim.
>
> | Quantile    | 0    | 0.2  | 0.4  | 0.6  | 0.8  | 1    |
> | ----------------- | ---- | ---- | ---- | ---- | ---- | ---- |
> | Proxy scores (VA) | 0.05 | 0.21 | 0.36 | 0.54 | 0.80 | 2.16 |
> | Oracle scores     | 0.00 | 0.19 | 0.38 | 0.61 | 0.89 | 1.79 |
>
>
> **3) Missing literature**
>
> Thank you for pointing out thoses interesting references. We will add the following paragraph to the paper:
>
> A significant body of literature already focuses on the problem of uncertainty quantification in ranking problems. In [1] for instance, the authors consider the Bradley-Terry-Luce (BTL) model and infer its general ranking properties. They show that, under this model, it is possible to control the False Discovery Rate (FDR) and thereby infer the top-K ranked items. In [2], the authors study the maximum likelihood estimator (MLE) and the spectral estimator to estimate the parameters of the BTL model, enabling them to construct confidence intervals for individual ranks. Similarly, [3] refines the analysis of [2] and shows that under a uniform sampling scheme, it is possible to efficiently estimate the preference scores of each item allowing them to construct simultaneous confidence intervals for the ranks of the items. Note that all these papers rely on the BTL model. This is a major difference  with our contribution as we do uncertainty quantification for any ranking algorithms and without relying on any specific underlying model.
>
> We hope we have addressed your concerns and are happy to discuss further if not.
>
> [1] Lagrangian Inference for Ranking Problems, Liu, Y., Fang, X., and Lu, J. (2023)
>
> [2] Uncertainty Quantification in the Bradley–Terry–Luce Model, Gao, G., Shen, Y., and Zhang, A. (2023)
>
> [3] Ranking Inference from Top Choices in Multiway Comparisons, Fan, J., Lou, Z., Wang, W., and Yu, M. (2025)
>
> [4] Asymptotics for conformal inference, Gazin (2024).

---

### Note · Authors · 2025-08-16

We would like to express our sincere gratitude to the reviewers, ACs, SACs, and PCs for their time, constructive feedback, and thoughtful discussions. Below, we provide a summary of our answers to the primary concerns raised by the reviewers.

**Set Size Consideration**

The large size of some sets arises from the intrinsic difficulty of the ranking task, rather than the use of the envelope. To demonstrate this, we show that our method closely approximates the performance of an oracle that does not need the envelope as it has access to the true ranks of the calibration points.

On the other hand, the sets constructed by the score function $s^{VA}$ have their size adapting to the ranking difficulty and are really tight for "easy to rank" points (see synthetic experiments of Fig. 4).

As suggested by reviewer 4soE,  we will add a comparison of the quantiles of the proxy scores and of the true scores and plot the length of the sets in function of the FCP.

**Feature Dependence**

As clarified in the rebuttal to reviewer eSy5, we emphasize that the constructed sets are inherently data-dependent.

**Conformal Nature**

We reaffirm the conformal nature of our method by highlighting that our coverage guarantee is distribution-free and that the concentration bound relies solely on the exchangeability assumption. The set sizes vary with the number of calibration points, but this is not contradictory to being conformal.

**Practical Setting**

Reviewer MSYV expressed concerns regarding the realism of the setting and how to use our method in the case where the true ranks of the calibration points are coming from a ranking algorithm, e.g., combining pairwise comparisons. This case completely matches our setting, as the true ranks can be defined as the output of a reliable algorithm, as done in the experiments. Then, our method provides intervals for the rank that this algorithm would have output. This is useful if this trustly algorithm is not runnable on test points (no available pairwise comparisons for them, too high computational cost...).

**Other Applications**

We will specify how to derive top $k$ sets and prediction sets for the ranks of only the test items from our method, along with the associated guarantees as discussed with reviewers C52t and MSYV.

**Missing Literature**

We acknowledge the absence of references to the BTL model and will include the paragraph written in our response to reviewer 4soE.

---

### Decision · Program_Chairs · 2025-09-17

**Decision:**

Accept (poster)

**Comment:**

This paper considers the problem of applying conformal prediction (CP) to provide uncertainty quantification of full ranking algorithms based on black-box prediction. The proposed CP method produces sets for the correct ranking for each element to achieve two coverage guarantees: marginal coverage and controlled false coverage proportion (important but less studied in the CP literature). Since the calibration data contains true ranking for only a subset of elements, the paper proposes methods to compute lower and upper bounds for the unknown groud-truth and how to use them form conformal calibration. Overall, this is a novel problem space for CP which is important in real-world applications and the paper is mostly well-executed.

Most reviewers' liked the paper but also asked a number of questions which were mostly addressed as part of the author response and discussion. The writing of the paper could be improved in terms of real-world motivation that matches with the problem formulation and experiments as per the Reviewer MSYV. The paper can also include some discussion on how to use the uncertainty sets in practice. These are minor concerns and I think can be addressed in the final paper (as per the authors' responses).

By considering all the factors, I'm leaning towards accepting the paper -- This is a novel problem space and will likely inspire the CP community to build on it. I strongly encourage the authors' to address the above two recommendations and incorporate all the discussion elements in the final paper to improve the clarity of the paper.